# A filtering reconfigurable intelligent surface for interference-free wireless communications

Jing Cheng Liang[1,2,5], Lei Zhang [1,2,5], Zhangjie Luo[1,2] ✉, Rui Zhe Jiang[1,2], Zhang Wen Cheng[1], Si Ran Wang [1,2], Meng Ke Sun[1,2], Shi Jin[3], Qiang Cheng [1,2,4] ✉ & Tie Jun Cui [1,2,4] ✉

The powerful capability of reconfigurable intelligent surfaces (RISs) in tailoring electromagnetic waves and fields has put them under the spotlight in wireless communications. However, the current designs are criticized due to their poor frequency selectivity, which hinders their applications in real-world scenarios where the spectrum is becoming increasingly congested. Here we propose a filtering RIS to feature sharp frequency-selecting and 2-bit phase-shifting properties. It permits the signals in a narrow bandwidth to transmit but rejects the out-of-band ones; meanwhile, the phase of the transmitted signals can be digitally controlled, enabling flexible manipulations of signal propagations. A prototype is designed, fabricated, and measured, and its high quality factor and phase-shifting characteristics are validated by scattering parameters and beam-steering phenomena. Further, we conduct a wireless communication experiment to illustrate the intriguing functions of the RIS. The filtering behavior enables the RIS to perform wireless signal manipulations with anti-interference ability, thus showing big potential to advance the development of next-generation wireless communications.

Reconfigurable intelligent surface (RIS) is also called programmable metasurface, which is a two-dimensional electromagnetic (EM) metamaterial integrated with tunable components and is controlled by digital modules such as field-programmable gate arrays (FPGAs) circuit[1–6]. It can dynamically and flexibly manipulate the properties of EM waves and fields in a programmable way, including amplitude, phase, polarization, and frequency, and thus it is naturally compatible with the information world[7–15]. Specifically, it can actively control wireless propagation and create an intelligent wireless environment. Together with other advantages like simple architecture, low cost, low power consumption, and easy deployment, RIS has attracted broad attention from the wireless community, and numerous theoretical innovations and prototype measurements have demonstrated the

broad applications in both 5 G and future 6 G networks, bringing a new paradigm to the future wireless communications[16–32]. Recent studies have demonstrated the immense potential of RIS in realistic deployment, propelling its value in practical applications to unprecedented heights[33,34].

RISs have brought about revolutionary changes to the wireless community in two aspects. The first is the RIS-based simplified-architecture wireless transmitters that directly realize signal modulations without using complicated digital-analog converters, mixers, or other devices in the conventional transmitting systems[23–26]. The other is the RIS-assisted wireless environment modulations, which means optimizing the communication quality by the powerful beam manipulation capabilities of the RIS[27–32]. Recently, the second application has

[1]State Key Laboratory of Millimeter Waves, Southeast University, Nanjing 210096, China. [2]Institute of Electromagnetic Space, Southeast University, Nanjing 210096, China. [3]National Mobile Communications Research Laboratory, Southeast University, Nanjing 210096, China. [4]Frontiers Science Center for Mobile Information Communication and Security, Southeast University, Nanjing 210096, China. [5]These authors contributed equally: Jing Cheng Liang, Lei Zhang. ✉e-mail: zjluogood@seu.edu.cn; qiangcheng@seu.edu.cn; tjcui@seu.edu.cn

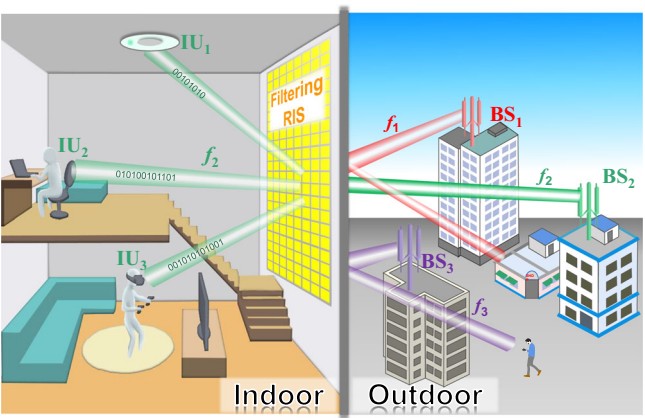

**Fig. 1 | A typical application scenario of the proposed filtering RIS, which is on a large scale and covers the entire wall of a shielded room.** The base stations, named $BS_1$, $BS_2$, and $BS_3$, are located outdoors working at three adjacent frequencies $f_1$, $f_2$, and $f_3$, respectively. The filtering RIS placed indoors on the wall aims to enhance the quality of the wireless communications between the base station $BS_2$ and the indoor users $IU_1$, $IU_2$, and $IU_3$ by generating specific pencil beams and collimating them accurately toward the targets. Different from the conventional RISs, the filtering RIS exhibits a powerful frequency-selecting ability that allows only the $f_2$ signal to enter the room, but strongly rejects the outdoor $f_1$ and $f_3$ signals. Hence, potential interference issues caused by the out-of-band signals can be eliminated.

garnered widespread interest, especially in the 5 G and next-generation mobile communications. It should be noticed that the practical wireless mobile communication scenarios often have multiple networks occupying closely adjacent spectra. However, existing RISs typically have broadband properties that not only tune signals in the target spectrum but also affect nontarget signals, resulting in serious network coexistence problems or even security concerns[35,36]. In addition, most of the current RISs failed to consider the impact of electromagnetic interference or noise, no matter if it is intentional or non-intentional[37]. This lack of frequency selectivity, which can be measured in terms of a quality factor (Q factor), hinders their practical deployment. This issue should be taken seriously in today's wireless environment with the increasingly crowded spectrum.

To tackle the above problems, we propose a filtering RIS that can selectively manipulate the wireless channels in a specific narrow frequency band and reject the signals out of the band. Figure 1 depicts an exemplary scenario to illustrate its typical applications. Three base stations, namely $BS_1$, $BS_2$, and $BS_3$, are located outside of the room and operate at three adjacent frequencies $f_1$, $f_2$, and $f_3$, respectively. A large-scale RIS is mounted on the wall indoors, aiming at enhancing the wireless communications between the indoor users ($IU_1$, $IU_2$, and $IU_3$) and $BS_2$ at the frequency $f_2$. This is realized by generating multiple narrow beams and collimating them precisely towards the indoor users. More significantly, the RIS possesses a powerful frequency-selecting feature to allow only the $f_2$ signals to enter the room but block out the $f_1$ and $f_3$ signals. As a result, the presence of the RIS ensures that the wireless communication indoors is not disturbed by signals from $BS_1$ and $BS_3$. We remark that the aim of the filtering RIS is different from the widely-used filters in user devices to enhance their performance; instead, they serve to actively regulate the spectrum within a confined environment without requiring a temporary refitting of all devices.

As a proof of concept, we design the RIS with advanced filtering and reconfigurable phase-shifting functionalities, which features exceptional capabilities in precisely selecting incoming signals in the frequency domain and provides flexible beamforming in the spatial domain, showing stronger power than its predecessors to modulate wireless environments. It is built based on the receiver-transmitter

metasurface structure for the sake of low profile and wide phase-shifting range with 2-bit modulations[38-42]. Notably, the integrated filtering modules endow the frequency-selective characteristic, which is described by an enhanced Q factor that surpasses most previous studies. Its central frequency is 3.5 GHz with a 200 MHz passband, which falls in the 4 G LTE Band 42 and the primary band for the 5 G technology (https://www.4g-lte.net/about/lte-frequency-bands/lte-band-42/, "5G Spectrum Public Policy Position" in white paper, 2017). An RIS prototype is fabricated and a series of numerical simulations and experiments are conducted for the property validations and application demonstrations. The transmission coefficients in the frequency domain are measured, showing the rejection rate of over 20 dB in the stopbands and reconfigurable 2-bit phase coding capabilities in the passband. Then, the RIS's ability to customize the wireless propagation environment is proved by manipulating far-field transmission patterns with steered EM beams. To further showcase the practical applications, a wireless communication experiment is carried out with the RIS placed between the transmitting and receiving modules. With the help of the RIS, the out-of-band signals are denied; only the signals in the passband can transmit through and be dynamically redirected to desired directions. By correctly setting the frequency of the system and the position of the receiving module, wireless connections are established through the RIS, and the information of color pictures is transferred and recovered successfully. The proposed RIS offers a promising solution to the issue of frequency interference that is not addressed in previous RIS-assisted wireless systems. It advances the practical deployment of RIS, particularly in complex propagation environments with congested spectrum. Compared with the conventional frequency selective surfaces (FSSs) that have been widely employed for frequency selectivity[43-47], the proposed single-layer RIS has the advantage of significantly improved filtering performance. Beyond that, the proposal's flexible beamforming capability makes it more useful for increasing the signal strength in target directions and enhancing the interference immunity for indoor wireless communications[30,34,48-50]. Detailed comparisons between the RISs and FSSs are provided in Supplementary Information Note 10. Unlike the conventional repeaters or relays that contain active components like analog-to-digital/digital-to-analog converters, mixers, and power amplifiers, the proposed RIS has significantly lower power consumption and complexity, and is free of additive noises[51]. More discussions on the RIS and conventional repeaters or relays are provided in Supplementary Information Note 11.

## Results
### Design of filtering RIS
Figure 2a, b show the two commonly used schematic frameworks for the conventional RISs: the stacked multilayer RIS[44,45,52-54] and receiver-transmitter RIS[38-42]. Based on the concept of frequency-selective-surface (FSS), the stacked multilayer RIS can exhibit satisfactory phase-shifting property performance, but the high profile makes it less preferred in wireless communication systems. Moreover, its transmission phase is shifted by adjusting the relatively wide instantaneous transmission band. As can be seen in Fig. 2a, the regulated phase-shifting behavior happens only in the narrow overlapping spectrum of the tunable band; the signals outside of this overlapping area but still within the instantaneous band are also allowed to be transmitted with random phase changes. The second type, receiver-transmitter RIS, modulates the phase in the operating band by using the transmission-line phase shifters or reversing the mode current. Compared with the first type, its profile is much lower, but its operating bandwidth is relatively wider, as illustrated in Fig. 2b. The two types of RIS have low Q factors, which means that much wider bandwidths are occupied than the frequency range with expected phase-shifting properties.

The working mechanism of the proposed transmission-type filtering RIS is plotted in Fig. 2c. It is based on the receiver-transmitter

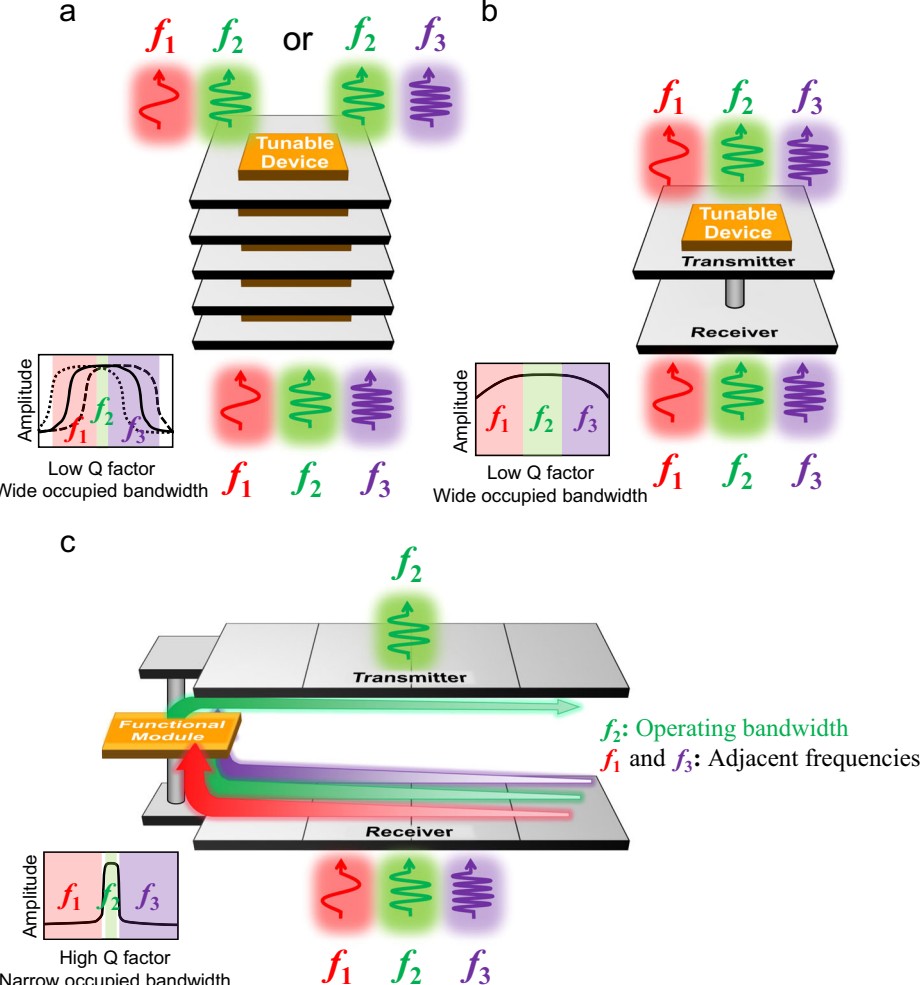

**Fig. 2 | Schematic diagrams and transmission amplitude spectra of three types of RISs. a** A unit cell of the conventional stacked multilayer RIS. **b** A unit cell of the conventional receiver-transmitter RIS. These two types of RIS have low Q factors and wide occupied bandwidths. **c** A subarray of the proposed filtering RIS with a high Q factor for strong frequency-selective property and a narrow occupied bandwidth. A functional module is integrated into the subarray.

RIS architecture, with a "functional module" inserted between them that allows the RIS to be extended with more complex functionalities and greatly enhances the freedom of RIS designs[55]. Here, a lumped filter and a phase shifter are integrated into the module. The high-Q filter provides a strong out-of-band rejection and low passband insertion loss, thus addressing the low-Q problem of the conventional receiver-transmitter RIS. The phase shifter enables reconfigurable modulations of the passband signals without degrading the filtering performance. By fully exploiting the advantages of each component, a large phase-shifting range in the passband and enhanced filtering performance can be achieved.

In practical application scenarios, the proposed filtering RIS would be implemented on a much larger scale, requiring a significant number of elements. In this study, we focus on a small panel that features 1 × 4 subarrays to demonstrate the feasibility of our proposed concept, as depicted in Fig. 3. Each subarray includes a receiver, a transmitter, and a functional module. Each receiver contains four parasitic rectangular patches and the microstrip lines below. The microstrip lines are specially designed such that the signals received by the four patches are in phase. The details of the module, which is composed of a phase shifter integrated with a filter chip, are shown in the upper left inset of Fig. 3. The phase shifter consists of a 0°/180° phase shifter cascaded with a 0°/90° phase shifter. Eight PIN diodes are embedded as the modulating components in the phase shifter. Several

metallic vias and narrow lines are connected to the microstrip through RF chokes, acting as the DC routes to bias the diodes. The output of the module is connected to the transmitter on the other side of the ground by a metallic through-via. Round clearances are located on the ground such that the through-vias do not touch it. The transmitter is located on the other side of the metallic ground, which shares exactly the same structure as the receiver. More details on the filter chip, the phase shifter, the receiver, and the transmitter can be found in Supplementary Information Notes 2, 3, and 4.

When the spatial waves impinge on the receiver, they are converted to guided waves, which are then injected into the functional module. Firstly, the signals are transmitted through the filter that offers a rejection of over 20 dB in the stopbands. It should be mentioned that the phase of the signals in the passband is not affected by the filter. Then the signals go into the phase shifter. By switching the ON or OFF states of PIN diodes, different lengths of microstrip paths are chosen, and thus the phase can be adjusted by four reconfigurable states with a 90° interval. After the phase-shifting, the signals are guided to the transmitter on the other side of the ground by the through-via and finally radiated into space. Three aspects should be mentioned. Firstly, the filters in the subarrays are the same, but the phase-shifting behaviors are independent. Hence, flexible manipulations of EM waves can be realized, such as beam-steering and more advanced multichannel communications. Secondly, the RIS is a

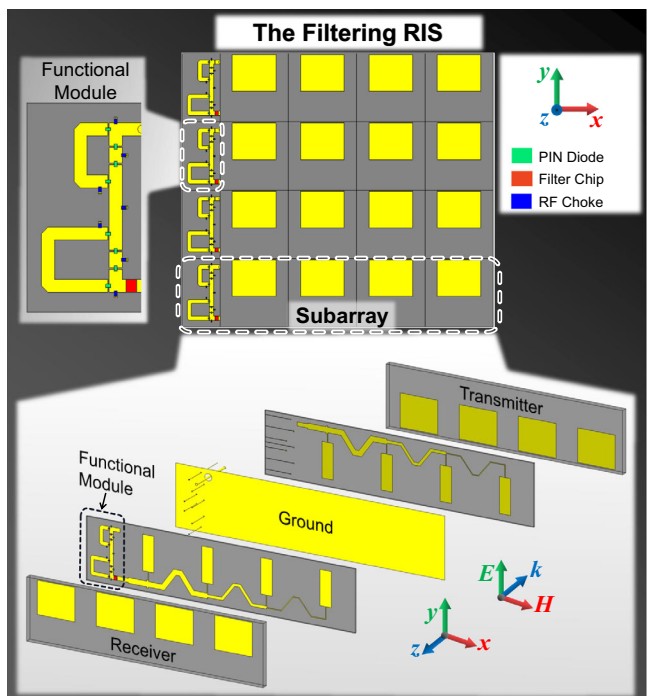

**Fig. 3 | Configuration of the proposed filtering RIS.** The details of a subarray are shown at the bottom, and the functional module is presented in the upper left inset.

reciprocal device, that is to say, the wave propagation through it is reversible when the roles of the receiver and transmitter are interchanged. These are important properties for channel modulations of wireless communications. Thirdly, although the filtering RIS is currently designed to support a single polarization, the polarization-independent properties can be practically realized by adopting the dual-polarized transmitter and receiver in the element[55,56].

To evaluate the performance of the filtering RIS, field-circuit cosimulations are performed using the commercial software CST Microwave Studio. An array of $1 \times 4$ subarrays is analyzed here, and the simulated transmission coefficients are shown in Fig. 4a, b, where four coding states (states 0, 1, 2, 3) are defined by the four transmission phase-shifting values with a 90° interval. The 2-bit phase coding states are switched by tuning the PIN diodes. From the transmission amplitude spectra in Fig. 4a, we observe that the four curves exhibit a significant overlap, suggesting the consistency of the amplitude-frequency responses of the RIS. An insertion loss of <2.5 dB is achieved in the passband from 3.4 to 3.6 GHz, along with a 20-dB rejection on the two sides of the passband. As the transmission amplitudes are extremely low in the stopbands, the phase response is meaningless. Therefore, the transmission phase spectra in the passband are presented in Fig. 4b. In the passband, stable 90° phase differences are exhibited between the curves, which enables a good 2-bit phase coding characteristic. The simulation results of the transmission coefficients demonstrate that the out-of-band signals are effectively suppressed, while the passband signals are transmitted with the stable 2-bit reconfigurable phase shifting.

To visually demonstrate the filtering capability of the RIS, Fig. 4c shows the electric-field (E-field) intensity distribution on the yoz-plane at 3.3–3.7 GHz when the EM waves normally impinge on the array. The propagation direction is along the z-axis, and the polarization is along the y-axis. The RIS is situated in the division between the upper and lower spaces in the figures, marked by a white dashed frame. The incident plane wave propagates downwards. At 3.4, 3.5, and 3.6 GHz, the EM waves can be transmitted to the lower half-space through the RIS. As all RIS elements are operated with the same transmission

amplitude and phase, the EM waves propagate vertically downwards on the yoz-plane. In contrast, the EM waves can hardly penetrate at 3.3 and 3.7 GHz, demonstrating the excellent rejection performance of the RIS in the stopbands. These full-wave simulations effectively illustrate the filtering functionality of the proposed RIS, allowing the EM waves in the passband to pass through while blocking those outside.

Table 1 provides a comparison of the filtering properties between the proposed RIS and the ones from refs. 41,42,44,45. Here the operating bandwidth (BW) means the frequency range with the acceptable phase-shifting ability; BWndB refers to the bandwidth in which the transmission amplitude is lower than the maximum value by less than $n$ dB. The ratio BW/BW3dB is used to evaluate the operating BW as a percentage of BW3dB. It can be observed that the BW/BW3dB of the stacked multilayer RIS is very small because the operating BW is much narrower than BW3dB[44,45,52–54]. Compared with them, the proposed RIS has a much larger BW/BW3dB value of 96%, suggesting a very narrow occupied bandwidth. Besides, the following two parameters are employed to quantitively measure the filtering property. The first one is the Q factor, which is defined by the ratio of the center frequency $f_0$ over BW3dB. The second one is the rectangle coefficient K20dB, which is defined by the ratio BW20dB/BW3dB. Normally, the rectangle coefficient K20dB is larger than 1; the ideal value of 1 means steep transitions on the two edges of the transmission curve, thus exhibiting a perfect filtering effect. Therefore, an ideal RIS should have a large Q factor and a K20dB close to 1. We are delighted to read from Table 1 that the Q factor is 14 in our design, which is much higher than the current studies; the K20dB is 1.3, suggesting a comparable value to the stacked multilayer RISs. The high rejection characteristics of the RIS outside its operating band, combined with its steep transition bands, can effectively avoid potential interferences caused by out-of-band signals.

By independently controlling the transmission phase of each subarray, beam-steering on the yoz-plane is enabled here. In the passband, the four subarrays of RIS are encoded with five sequences ("3210", "1100", "0000", "0011", and "0123", where "0, 1, 2, 3" abbreviate the digital states 0–3) for examples. The simulated far-field patterns of the transmitted waves at 3.5 GHz are plotted in Fig. 4d, showing that the transmitted beams are steered to the directions with elevation angles of −28°, −12°, 0°, 12°, and 28°, respectively. These patterns agree quite well with the theoretical results presented in Fig. 4e, which are calculated by ref. 1

$$F(\theta,\varphi)=E_{m,n}(\theta,\varphi)\sum_{m=1}^{M}\sum_{n=1}^{N}\exp\{-i\{\varPhi_{m,n}+kd\ \sin\theta[(m-1/2)\cos\varphi$$
$$+(n-1/2)\sin\varphi]\}\}$$

(1)

where $M$ and $N$ are numbers of elements in the x- and y-directions; $E_{m,n}(\theta,\varphi)$ is the transmission far-field pattern of the element $(m, n)$; $\varPhi_{m,n}$ is the transmission phase of the element $(m, n)$; $k$ is the wavenumber of the EM wave in free space; $d$ is the period of the receiver and transmitter elements; $\theta$ and $\varphi$ are the elevation and azimuthal angles, respectively.

To further illustrate the anomalous transmission effects, Fig. 4f presents the simulated near-field spatial distributions of the E-field intensities on the yoz-plane at 3.5 GHz when the RIS is controlled by the five coding sequences. It is clearly observed that the plane waves propagate along the -z direction before they interact with the RIS; after they pass through, they are deflected off the normal direction with the specific angles, which are determined by the coding sequences, that is, the phase distributions on the RIS.

The full-wave simulations effectively illustrate the beam manipulation capabilities of the proposed RIS, which can dynamically steer the passband waves towards the specified directions by encoding the RIS. This highlights its potential to control the wireless channels and

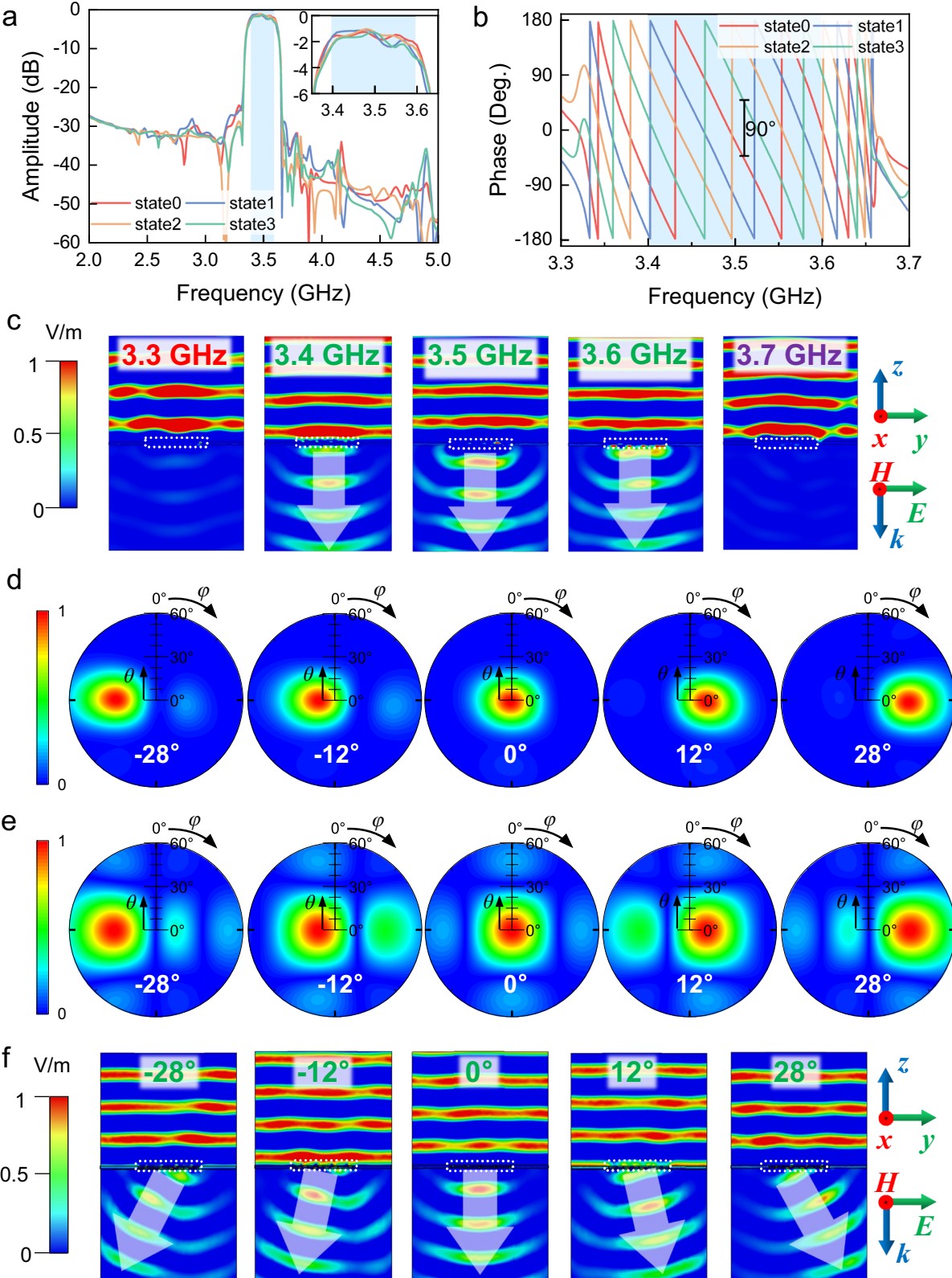

**Fig. 4 | The transmission coefficients of the RIS obtained from the cosimulations. a** Amplitude and (**b**) phase spectra of the four digital states (States 0–3). A stable 90° phase difference between adjacent states is achieved in the passband. **c** The *E*-field intensity distributions on the *yoz*-plane. The position of RIS is marked by the white dashed frame. **d** Simulated and (**e**) theoretical 2D far-field patterns of the filtering RIS with the beam scanning angles of −28°, −12°, 0°, 12°, and 28° at 3.5 GHz. **f** The *E*-field intensity distributions on the *yoz*-plane when the anomalous transmissions happen at 3.5 GHz. The position of the RIS is marked by the white dashed frame.

**Table 1 | Comparison between this work and the conventional RISs**

| Ref. | Type | $f_0$ (GHz) | Operating BW (GHz) | BW3dB (GHz) | BW20dB (GHz) | BW/BW3dB | Q factor | K20dB |
|------|------|-----------|---------------------|-------------|--------------|----------|----------|-------|
| This work | Rx. - Fun. - Tx. | 3.5 | 0.24 (6.9%) | 0.25 (7.1%) | 0.32 (9.1%) | 96% | 14 | 1.3 |
| mmWave RIS (Simulated) | Rx. - Fun. - Tx. | 28.5 | 3.6 (12.6%) | 4.0 (14.0%) | 5.6 (19.6%) | 90.0% | 7.1 | 1.4 |
| [41] | Rx. - Tx. | 5.0 | 0.5 (10.0%) | 1.0 (20.0%) | 2.0 (40.0%) | 50.0% | 5.0 | >2.0 |
| [42] | Rx. - Tx. | 32.0 | 4.3 (13.4%) | 4.9 (15.3%) | 10 (31.2%) | 87.8% | 6.5 | 2.0 |
| [46] | Stacked 5-Layers | 5.4 | 0.2 (3.7%) | 0.9 (16.7%) | 1.1 (20.4%) | 22.2% | 6.0 | 1.2 |
| [46] | Stacked 5-Layers | 5.4 | 0.1 (1.9%) | 0.7 (13.0%) | 0.9 (16.7%) | 14.3% | 7.7 | 1.3 |

$f_0$, center frequency; BW bandwidth; BW$n$dB, $n$ dB bandwidth; Q factor, quality factor, the ratio $f_0$/BW3dB; K20dB, rectangle coefficient, the ratio BW20dB/BW3dB; Rx., receiver; Fun., functional module; Tx., transmitter.

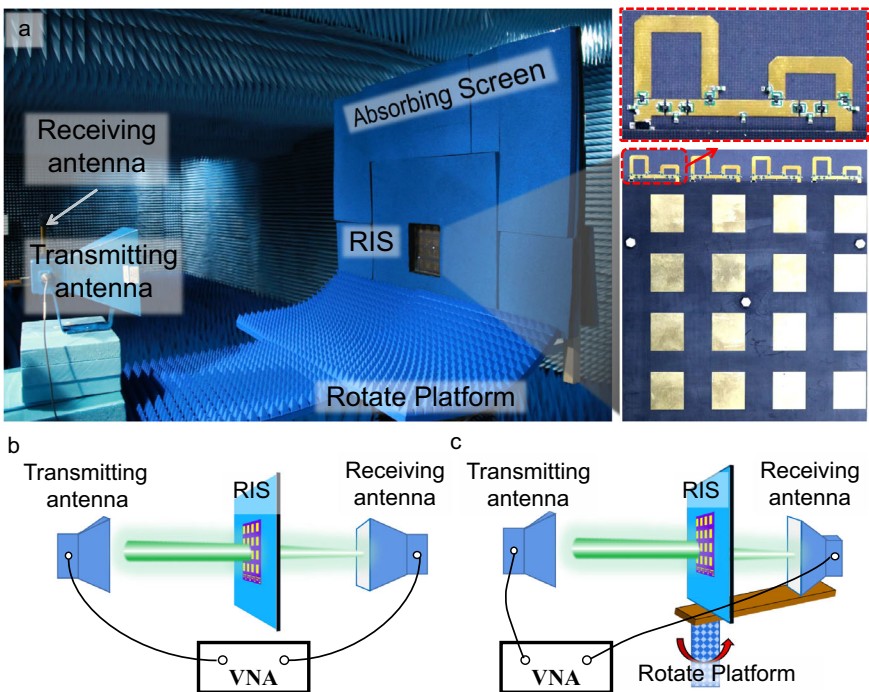

**Fig. 5 | Experimental setups. a** Photograph of the experimental environment in the microwave anechoic chamber. The pictures of the prototype are given in the insets. **b** The measurement setup for the transmission coefficient. **c** The measurement setup for the far-field patterns.

enhance the wireless communication service quality. It is important to note that only four subarrays are used in this study as a proof of concept to demonstrate the functions of RIS. The elements are controlled in columns, which is consistent with most of the previous studies[22,32–34,50,57–59]. Based on the validated mechanism, it is feasible to design a filtering RIS with a pair of transmitter and receiver and a functional module for filtering and phase-shifting, which can realize the two-dimensional (2D) beam-steering performance. In Supplementary Information Note 7, the design of the element and its simulated properties are presented. Through simulations, the 2D beamforming is also verified by using an array with 10×10 elements, showing the wide scanning ranges of ±63° and ±64° on the *xoz*- and *yoz*-planes, respectively. Additionally, the proposed concept can be moved to higher frequencies by adjusting the structural parameters accordingly and selecting an appropriate filter chip that operates at the desired target frequencies. In Supplementary Information Note 8, we have designed a filtering RIS that operates at the millimeter-wave (mmWave) frequencies using the same concept. Its filtering and phase-shifting properties are studied through field-circuit cosimulations. As shown in Table I, its performances are competitive when compared to the state-of-the-art work in the mmWave bands.

## Fabrication and measurement

An array of $1 \times 4$ RIS subarrays is fabricated using the printed circuit board technology. A series of experiments are performed to validate the performance of filtering, phase-shifting, and beam manipulation of the transmitted waves. Moreover, its advanced ability to re-arrange the wireless environment is vividly demonstrated by a real communication experiment. The picture of the prototype is shown in the right inset of Fig. 5a. It measures 160 mm × 182 mm, and the total thickness is 8.1 mm, or $0.09\lambda_0$, where $\lambda_0$ is the wavelength at 3.5 GHz. A digital controlling circuit board is designed and fabricated to provide the operating voltages for the PIN diodes in the phase shifters.

The transmission coefficients and far-field patterns of the proposed RIS are measured in a microwave anechoic chamber. Figure 6a displays the measured amplitude spectra of transmission coefficients under the normal incidence, with the passband 3.4–3.6 GHz denoted by the blue areas. The amplitudes of the four coding states within the passband range from −1.2 to −4.1 dB, with a rapid roll-off and a 20-dB rejection in the stopbands. The loss is mainly due to the insertion loss of the phase shifters and filter chips, which comes at the cost of phase shifting and sharp frequency selection features. More details about the

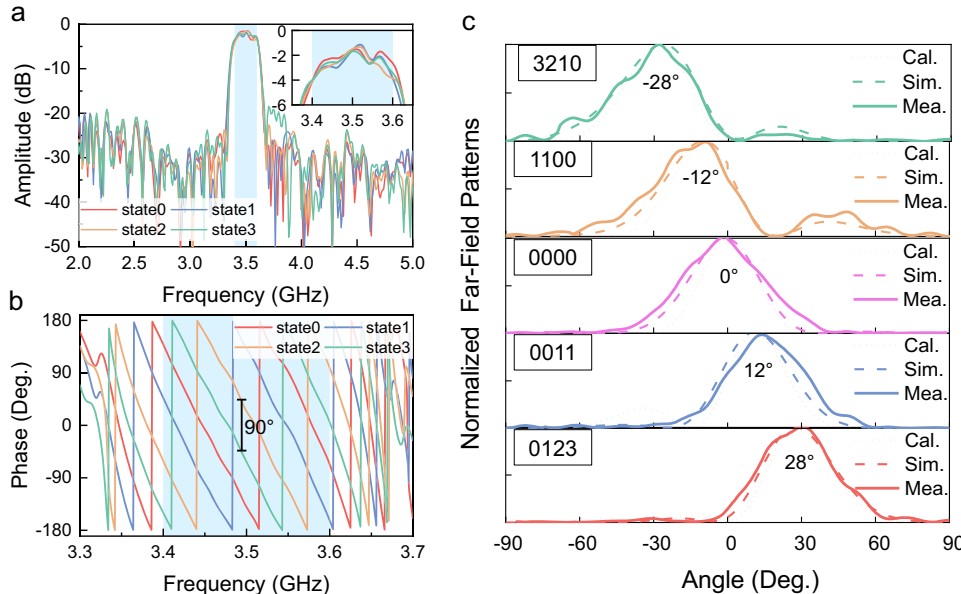

**Fig. 6 | The measurement results of the proposed RIS.** The transmission amplitudes (**a**) and phases (**b**) as functions of frequency for the four digital states (States 0–3). **c** The calculated, simulated, and measured far-field transmission patterns on the *yoz*-plane of the filtering RIS with the beam scanning to the angles −28°, −12°, 0°, 12°, and 28° at 3.5 GHz.

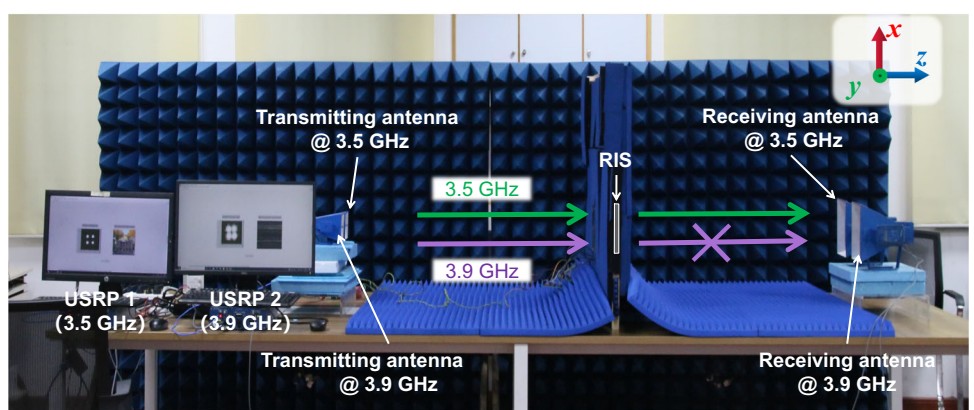

**Fig. 7 | Configuration of the wireless communication experiment.** A windowed absorbing screen is placed between the transmitting and receiving antennas, and the RIS is embedded in the window.

loss analysis are presented in Supplementary Information Note 5. To address this issue, low-noise amplifiers could be integrated into the functional modules to compensate for the attenuation. A 2-bit phase-shifting operation with the four reconfigurable coding states is realized within the passband, as shown in Fig. 6b. Both transmission amplitude and phase spectra agree well with the simulations. The performances under oblique incidences are discussed in detail in Supplementary Information Note 6.

Using the same experimental setup, the far-field transmission properties are measured. Figure 6c shows the results on the *yoz*-plane at 3.5 GHz under the control of the five coding sequences. The transmitted beam is deflected to −28°, −13°, 0°, 13°, and 27°, respectively, almost in consistence with the simulated and theoretical ones (−28°, −12°, 0°, 12°, and 28°), with an error of fewer than 2 degrees. The slight discrepancies can be attributed to fabrication and testing errors. Considering the wide beamwidth (half-power beamwidth of about 36°), it can still be claimed that the transmitted beams are indeed deflected to the expected angles controlled by the coding sequences on the RIS. The measured results demonstrate the effective filtering,

phase-shifting, and beam-steering abilities of the designed RIS, providing a solid hardware foundation for further applications in real wireless communication scenarios.

The wireless communication configuration and measured results are presented in Figs. 7 and 8, respectively. Two software-defined radio reconfigurable devices (USRP-2974, National Instruments Corp.)[25–27] that are set to work at 3.5 and 3.9 GHz, respectively, can encode a color picture into a binary stream and modulate it using a quadrature phase shift keying (QPSK) scheme. Two transmitting antennas and two receiving antennas are connected to the output and input of the two USRPs, respectively. Horn antennas with a working bandwidth that covers the entire frequency band of interest (3.1–3.9 GHz) are employed here to eliminate the uncontrollable multipath effects. A windowed absorbing screen is placed between the transmitting and receiving antennas, and the RIS is embedded in the window. The transmitting antennas are placed facing the RIS in a normal orientation, and the distance between the antennas and RIS is 1 meter. The positions of the receiving antennas are varied on the *yoz*-plane. The QPSK signals generated by USRPs are radiated by the antennas and

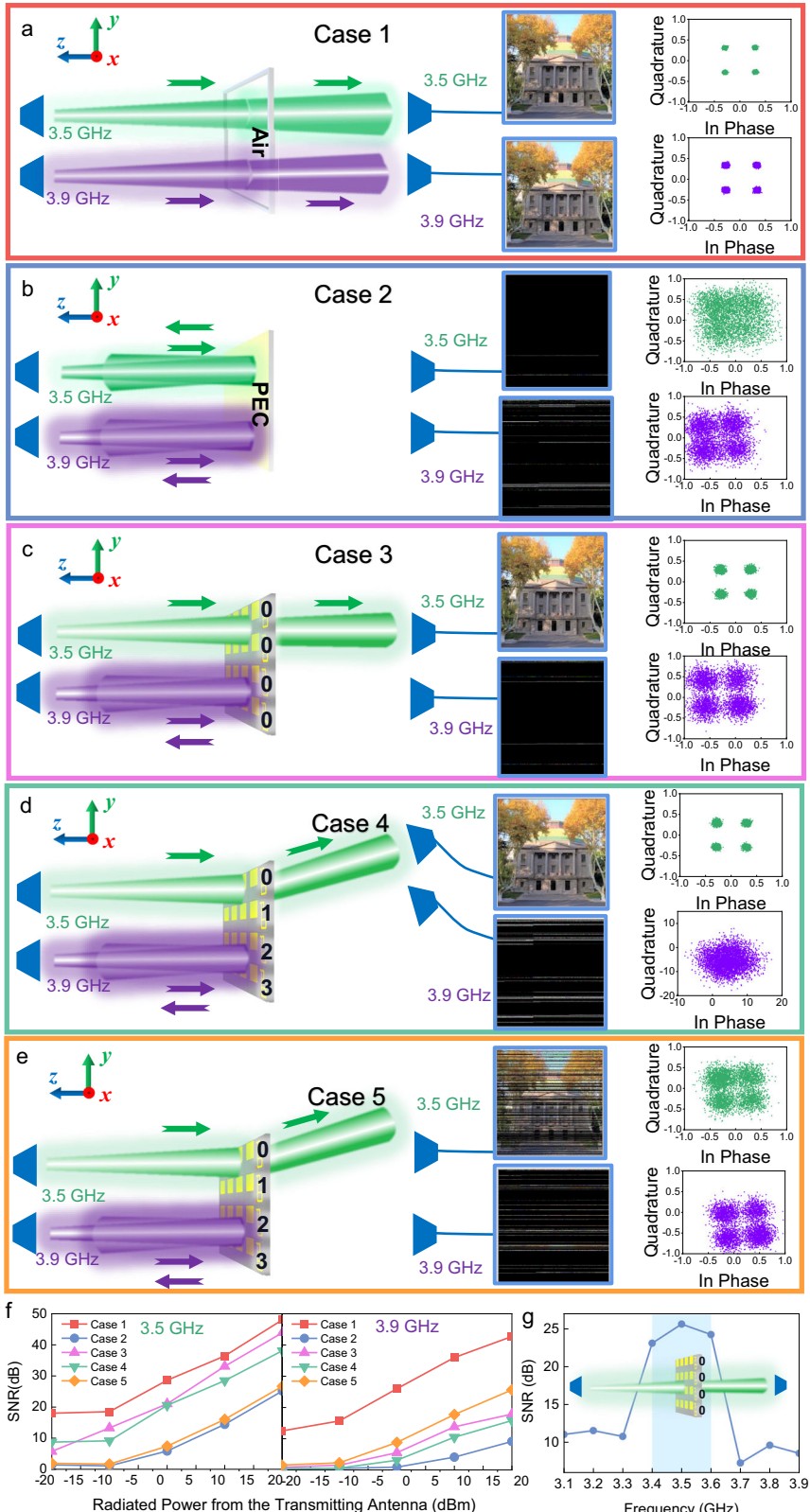

**Fig. 8 | The wireless communication experiments to demonstrate the proposed selectivity in both frequency and spatial domains of the filtering RIS. a–e** Five cases with different frequencies and coding sequences. **f** The SNR spectra in relation to the radiated power from the transmitting antennas in the five cases. **g** The SNR spectrum in Case 3 with the radiated power of 0 dBm.

**Table 2 | Settings of the RIS and receiving antennas in the wireless communication experiments**

| Cases | Coding sequences on the RIS | Theoretical beam direction | Receiving antenna direction |
|---|---|---|---|
| 1 | Air | – | 0° |
| 2 | PEC | – | All |
| 3 | 0000 | 0° | 0° |
| 4 | 0123 | 28° | 28° |
| 5 | 0123 | 28° | Any except 28° |

pass through the RIS before being received by the other set of antennas. The received signals are then demodulated by the USRPs to recover the pictures. The symbol rate in experiments is set as 200 KBaud, which is enough for picture transmissions.

Five different cases are designed to showcase the effectiveness of the filtering RIS, as shown in Fig. 8a–e. The radiated power from the transmitting antennas is 0 dBm. Settings of the RIS and directions of the receiving antennas are summarized in Table 2. Photographs of the measurements can be found in Supplementary Information Note 9. Cases 1 and 2 serve as the control groups. In Case 1, the absorption window with the same size as the RIS is left empty, while in Case 2, a metallic plate is placed inside it. As depicted in Fig. 8a, the demodulated constellation diagrams at both 3.5 GHz and 3.9 GHz are of high quality, and the pictures are satisfactorily recovered, implying that the signals are well received by the antenna through the window. In Case 2, on the contrary, the presence of a metal plate prevents the transmission, as shown in Fig. 8b. In Case 3, the metallic plate is replaced by the RIS, and the coding sequence on it is set to "0000", indicating that the passband signal (3.5 GHz) should be transmitted through the RIS in a direction perpendicular to the surface. The receiving antennas are placed in the correct direction. As shown in Fig. 8c, the demodulated constellation diagram at 3.5 GHz is of good quality, and the picture is satisfactorily restored, implying that the signal is well received. In Case 4, the coding sequence is changed to "0123", directing the beam to a 28° angle on the *yoz*-plane. By repositioning the 3.5-GHz receiving antenna to the correct direction, almost the same satisfactory results are observed, as presented in Fig. 8d. However, if the 3.5-GHz receiving antenna deviates from the expected direction (Case 5), the signal is no longer correctly received, as proved by the cluttered constellation diagram and the unrecovered picture in Fig. 8e. The experiments vividly demonstrate the wave-manipulation capability of the RIS in the passband. In sharp contrast, for the out-of-passband signal (3.9 GHz), regardless of the coding sequence or the receiving antenna's location on the right side, the picture cannot be restored, and the constellation diagram is cluttered. This strongly suggests that such signals are rejected by the RIS.

To further quantitatively evaluate the performance of the signal transmissions, we calculate the signal-to-noise ratio (SNR) spectrum in relation to the radiated power from the transmitting antenna in the five cases, as shown in Fig. 8f. It is observed that the SNR progressively increases with the rise in the radiated power, indicating an enhancement in transmission through the RIS. For the passband signal (3.5 GHz), the SNRs in Cases 1, 3, and 4 are significantly higher than those in Cases 2 and 5, proving the beam-steering capability of the RIS in the passband. For the out-of-band signal (3.9 GHz), the SNRs in Cases 2 through 5 are considerably lower than those in Case 1, suggesting the blocking effect of the RIS in the stopband. Fig. 8g shows the SNR spectrum in Case 3 using the horn antennas when the coding sequence is "0000" and the radiated power is 0 dBm. It can be seen that the maximum value of 25.7 dB occurs at 3.5 GHz. The value

exceeds 23.2 dB from 3.4 to 3.6 GHz, and it drops sharply below 11.6 dB outside the 3.3 to 3.7 GHz range, indicating a frequency window that allows the signals to be transmitted through the RIS efficiently. These results demonstrate the wave-manipulation capability of the RIS in the passband and interference mitigation for adjacent frequencies, aligning with the results presented in Fig. 8a–e.

Two additional wireless communication experiments are conducted to further illustrate the properties of the RIS. The first experiment is carried out by using the custom-built patch antennas as both transmitting and receiving antennas, and the second experiment is conducted outdoors by replacing the absorbing screen around the RIS with a brick wall. The photographs and results are given in Supplementary Information Note 9.

## Discussion

We propose a novel RIS that combines the filtering, phase-shifting, and beam-steering functions to assist wireless communications in a target channel and resist interferences from closely adjacent spectra. The presented interference-free characteristic is achieved by the strong frequency selectivity that distinguishes it from conventional ones, which allows only the signals in a specified narrow frequency window while rejecting the out-of-band signals. By adopting the receiver-transmitter structure integrated with the high-Q and phase-shifting functional modules, a RIS prototype is fabricated, and its performances of transmission coefficients and far-field patterns are measured in an anechoic chamber, which are consistent with the theoretical and simulated anticipations. The measured results show a rejection of over 20 dB in the stopbands and a Q factor of 14, a superior frequency-selecting feature that outperforms the current RIS designs. Moreover, the reconfigurable 2-bit phase coding property in the passband enables the dynamic manipulations of the wave propagation, which is validated by the measured far-field patterns. Thereafter, we demonstrate the filtering and beam-controlling effects of the proposed RIS in the wireless communication scenario, where the transmission of color pictures is restricted to the pre-set spatial direction and frequency band. It should be mentioned that beyond this work, the functionality can be further extended, such as the amplitude control, which can be expected if active amplifiers are integrated. The proposed filtering RIS shows impressive anti-interference and beam-controlling capabilities; thus, we believe it can find applications for wireless communications in congested frequency spectra and propagation environments.

## Methods

### Full-wave simulations

The full-wave simulations in this work were performed using commercial software, CST Microwave Studio 2019. An array of $1 \times 4$ subarrays were considered. The filter chips and PIN diodes in the array were modeled as equivalent RLC circuits, which are described in detail in Supplementary Information Note 3. Frequency solver and open boundaries were set in the simulations so that the RIS was illuminated by a uniform plane wave polarized along the *y*-axis. Absorbing materials ($\varepsilon_r = 2.78$, $\mu_r = 2.8$, tan $\delta_e = 2.47$, tan $\delta_m = 2.45$) were enveloped around the RIS to mitigate the mutual interference between the diffracted waves and transmitted waves. In addition, the electric field monitor and the far-field monitor were deployed to obtain the energy distribution in the near-field and far-field regions, respectively.

### Field-circuit cosimulations

The cosimulations were performed based on full-wave and circuit simulations. Discrete ports were set for the filter chip and the diodes in the 3D models in the full-wave simulations. After the full-wave simulations, the scattering parameter (S-parameter) file was imported to the circuit simulation environment, where

the discrete ports were connected to the filter chip and the diodes. Two microwave sources were connected to the two external ports, respectively, to excite the whole model. To calculate the transmission coefficients, the signals transmitted through the RIS were normalized by the signal transmitted through the same aperture when the RIS was removed.

**Measurement setup**

The transmission coefficients and far-field properties of the proposed RIS were measured in a microwave anechoic chamber. The setup is shown in Fig. 5b, c. Two linearly polarized horn antennas were utilized as the transmitting antenna and the receiving antenna, respectively. The transmitting antenna was placed 1.2 meters away from the RIS, connected to port 1 of a vector network analyzer (VNA) (Agilent N5245A). The receiving antenna and the RIS were placed on a rotating platform, and their distance was 9.5 meters. The receiving horn was connected to port 2 of the VNA. To reduce the diffraction around the RIS, a windowed absorbing screen was constructed, and the RIS was embedded within the window. The screen was covered with absorbing foams backed by a metal plate.

For the transmission coefficient measurement, the two horn antennas were placed on the two sides of RIS, oriented in the normal direction. For starters, the reference transmission signals were measured without the RIS. After that, the transmission signals with the presence of the RIS were measured. The RIS's transmission coefficients were obtained by normalizing the second data with the reference data. For the far-field transmission pattern measurement, the platform rotated with a mechanical turntable from -90° to +90° at increments of 1°. The transmission signals were recorded and normalized by the maximum amplitude, and the far-field patterns were finally plotted.

## Data availability

The authors declare that all relevant data are available in the paper and its Supplementary Information Files, or from the corresponding author on request.

## Code availability

The custom computer codes utilized during the current study are available from the corresponding authors on request.

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

## Acknowledgements

This work is supported by the National Natural Science Foundation of China (62288101, T.J.C. and Q.C., 62101123, L.Z., 61801117, Z.L.), the National Science Foundation (NSFC) for Distinguished Young Scholars of China (62225108, Q.C.), the National Key Research and Development Program of China (2018YFA0701904, Q.C., 2021YFA1401002, L.Z., 2023YFB3811504, L.Z.), the Program of Song Shan Laboratory (Included in the management of Major Science and Technology Program of Henan Province) (221100211300-02, Q.C.), the 111 Project (111-2-05, T.J.C.), the Jiangsu Province Frontier Leading Technology Basic Research Project (BK20212002, T.J.C.), the Natural Science Foundation of Jiangsu Province (BK20221209, Z.L.), the Fundamental Research Funds for the Central Universities (2242022k6003, Q.C., 2242023K5002, L.Z.), the Southeast University - China Mobile Research Institute Joint Innovation Center (R202111101112JZC02, Q.C.), and the National Postdoctoral Program for Innovative Talents (BX2021062, L.Z.), and the Young Elite Scientists Sponsorship Program by CAST (2020QNRC001, L.Z.).

Received: ((will be filled in by the editorial staff))
Revised: ((will be filled in by the editorial staff))
Published online: ((will be filled in by the editorial staff))

## Author contributions

J.C.L., L.Z. and Z.L. conducted the theoretical analysis, modeling, and numerical simulations. J.C.L., L.Z. and Z.L. wrote the paper. J.C.L. and Q.C. proposed the concept of the filtering RIS. J.C.L., R.Z.J., S.R.W. and M.K.S. built the RIS-assisted wireless communication experiment. J.C.L., R.Z.J., Z.W.C. and M.K.S. conducted experiments and data processing. T.J.C., Q.C. and S. J. provided suggestions and comments and helped to organize and revise the draft. All authors discussed the results and contributed to the manuscript.

## Competing interests

The authors declare no competing interests.
