## [Peer Review File · Nature Communications]

A filtering reconfigurable intelligent surface for interference-free wireless communicationsREVIEWER COMMENTS

Reviewer #1 (Remarks to the Author):

The paper presents a novel work on solving the problem of frequency selectivity in RIS which was a hinderance to its practical applicability. Authors proposed a filtering RIS to feature sharp frequency-selecting and bit phase-shifting properties, which permits the signals in a narrow target bandwidth to pass through and rejects the out-of-band signals. Experimental verification has been performed by fabricating the prototype and testing it in various environments.

Some minor comments are:

add some quantitative results on the %age of band pass and stop capabilities of the file.

Some important references are missing Usman, et al. Intelligent wireless walls for contactless in-home monitoring. Light Sci Appl 11, 212 (2022). <https://doi.org/10.1038/s41377-022-00906-5>

High-resolution programmable scattering for wireless coverage enhancement: an indoor field trial campaign. IEEE Transactions on Antennas and Propagation, 71(1), pp. 518-530. (doi:

10.1109/TAP.2022.3216555)

Reviewer #2 (Remarks to the Author):

The submitted manuscript describes a simultaneous transmitting reflecting intelligent surface that offers frequency selectivity. The paper is well written and can be of interest to scientific community. However, before its publication, authors need to make major changes.

1) It is not clear what is the advantage of the proposed device over the case where the filter chip is placed in the signal path of the users? In fact, most of them come with such filters. They are super low cost, no power consumption, and low weight.

2) The choice of frequency is also not clear. Such low frequencies can pass through walls. So even if a small 4 by 4 RIS rejects the interfering signals, the interfering signals can pass through the walls and reach the users. So it is not really clear what is the advantage here. This is very important because the authors compare their work with other works in Table 1 which are at higher frequencies over which EM signals have smaller penetration depths and implementing filters with higher Q are more difficult.

3) Please add a comment on how this device can be extended to 2D? It does not seem to be straightforward as the signal collected by each subarray is phased differently (in contrast to conventional RISs where each element is phased). This is important because figure 1 promises a 2D beamforming capabilities by the implemented device does not yield that.

4) Please provide a reference for the following sentence: "...however, their practical deployments are hindered by the limited frequency selectivity, which is described quantitatively by the quality factor (Q factor)...." To the best of my knowledge, that has not been a hindering factor. There are obstacles in using RISs in practice, but their low Q has not been an issues.

5) What is the difference between this device and a repeater/relay? All seems to be the same except that repeaters/relays allow for amplification of the signal which is in contrast to the 50% loss of power offered by this device.

6) I believe the proposed communication demonstration does not show interference mitigation which is the promise of the work. Instead of using absorbers around the device, the authors should use typical walls. On one side, place two sources, one with 3.5 GHz and one with 3.9 GHz. See the received signal on the other side for both and report SNIR with and without RIS.

7) Can this work be used at higher frequencies? If yes, how can the losses be kept at minimum? The 50% loss of power in this work is not small.

8) Please comment on what communication link in practice uses horn antennas for communication at 3.5 GHz band? This is a very important question because spatial selectivity of high gain antennas already provides with some level of interference mitigation.

9) It is not clear why authors refer to generalized Snell's law in discussion for (1). This is a typical receiving/transmitting antenna. Its operation can be described using antenna array theory.

RE: Manuscript No. NCOMMS-23-39119, “A filtering reconfigurable intelligent surface for interference-free wireless communications”

RESPONSES TO REVIEWERS’ COMMENTS

The authors would like to thank all editors and reviewers for their valuable comments and suggestions in enhancing the quality of this manuscript. We have taken all comments seriously and made every effort to respond point by point. All corresponding changes are highlighted in the revised manuscript. We sincerely hope that the comments and suggestions have been addressed adequately.

Reviewer #1:

Comment:

The paper presents a novel work on solving the problem of frequency selectivity in RIS which was a hinderance to its practical applicability. Authors proposed a filtering RIS to feature sharp frequency-selecting and bit phase-shifting properties, which permits the signals in a narrow target bandwidth to pass through and rejects the out-of-band signals. Experimental verification has been performed by fabricating the prototype and testing it in various environments.

Response:

Thank you for the positive feedback on the manuscript. It means a lot to us and is a great source of encouragement.

Comment:

Some minor comments are:

1. add some quantitative results on the %age of band pass and stop capabilities of the file.

Response:

Thank you very much for the kind suggestion. We have added %age results to the parameters BW3dB and BW20dB in Table 1 to describe the bandpass and stop capabilities of the proposal. BW n dB refers to the bandwidth in which the transmission amplitude is lower than the maximum value by less than n dB.

Corresponding change:

Table 1. Comparison between this work and the conventional RISs.

Ref.	Type	f_0 (GHz)	Operating BW (GHz)	BW3dB (GHz)	BW20dB (GHz)	BW/BW3dB	Q factor	K20dB
This work	Rx. - Fun. - Tx.	3.5	0.24 (6.9%)	0.25 (7.1%)	0.32 (9.1%)	96%	14	1.3
mmWave RIS (Simulated)	Rx. - Fun. - Tx.	28.5	3.6 (12.6%)	4.0 (14.0%)	5.6 (19.6%)	90.0%	7.1	1.4
[41]	Rx. - Tx.	5.0	0.5 (10.0%)	1.0 (20.0%)	2.0 (40.0%)	50.0%	5.0	>2.0
[42]	Rx. - Tx.	32.0	4.3 (13.4%)	4.9 (15.3%)	10 (31.2%)	87.8%	6.5	2.0
[46]	Stacked 5-Layers	5.4	0.2 (3.7%)	0.9 (16.7%)	1.1 (20.4%)	22.2%	6.0	1.2
[47]	Stacked 5-Layers	5.4	0.1 (1.9%)	0.7 (13.0%)	0.9 (16.7%)	14.3%	7.7	1.3

f_0 , center frequency; BW, bandwidth; BW_n dB, n dB bandwidth; Q factor, quality factor, the ratio f_0/BW_{3dB} ; K20dB, rectangle coefficient, the ratio BW_{20dB}/BW_{3dB} ; Rx., receiver; Fun., functional module; Tx., transmitter.

Comment:

2. Some important references are missing Usman, et al. Intelligent wireless walls for contactless in-home monitoring. Light Sci Appl 11, 212 (2022). <https://doi.org/10.1038/s41377-022-00906-5>, High-resolution programmable scattering for wireless coverage enhancement: an indoor field trial campaign. IEEE Transactions on Antennas and Propagation, 71(1), pp. 518-530. (doi: 10.1109/TAP.2022.3216555)

Response:

Thank you very much for your insightful suggestion. The concept of intelligent wireless walls, proposed in Ref. [1], ensured high-precision activity monitoring in complex environments wherein conventional microwave sensing is invalid. Furthermore, Ref. [2] presented an investigation of field trials in realistic indoor deployments, ascertaining the coverage enhancement performance for three common wireless communication scenarios. These significant works explored the potential of reconfigurable intelligent surfaces (RISs) in realistic deployments. Therefore, we have added these two references in the revised manuscript.

Corresponding change:

In the first paragraph of the “Introduction” section, these two significant works are referenced: “Recent studies have demonstrated immense potentials of RIS in realistic deployment, propelling its value in practical applications to the unprecedented heights^{33, 34}.”

33. Usman, M. et al. Intelligent wireless walls for contactless in-home monitoring. *Light Sci. Appl.* **11**, 212 (2022).

34. Rains, J. High-Resolution Programmable Scattering for Wireless Coverage Enhancement: An Indoor Field Trial Campaign. *IEEE Trans. Antennas Propag.* **71**, 518–530 (2023).

Reviewer #2:

Comment:

The submitted manuscript describes a simultaneous transmitting reflecting intelligent surface that offers frequency selectivity. The paper is well written and can be of interest to scientific community. However, before its publication, authors need to make major changes.

Response:

We appreciate the positive feedback from the reviewer. Their comments are helpful in refining the manuscript. We have taken them seriously and addressed each point accordingly.

Comment:

1) It is not clear what is the advantage of the proposed device over the case where the filter chip is placed in the signal path of the users? In fact, most of them come with such filters. They are super low cost, no power consumption, and low weight.

Response:

Thank you very much for the comment. It is indeed important to emphasize the advantages of the proposal more prominently in the manuscript.

We agree with the reviewer that filters are commonly used in user devices to eliminate unwanted frequencies and permit the right ones to help improve the performance of the devices. In this work, on the other hand, the aim is to actively regulate the spectrum of a confined environment by integrating the filtering functions into the metasurface. For example, in certain scenarios, specific frequencies may need to be blocked for security reasons. The proposed filtering RIS can serve this purpose without requiring a temporary refitting of all devices. The filtering RIS and the filters embedded in devices are not interchangeable with each other.

Furthermore, the advantages of the proposed RIS also lie in the flexible beamforming capability. By customizing the transmission phase distribution on the metasurface, we can intentionally direct signals in specific directions within the passband. This can be utilized to modulate the wireless environment and enhance signal strengths in the spatial domain.

Corresponding change:

At the end of the third paragraph of the “Introduction” section, the difference between the commonly used filters in devices and the proposed filtering RIS is explained.

“We remark that the aim of the filtering RIS is different from the widely-used filters in user devices to enhance their performance; instead, they serve to actively regulate the spectrum within a confined environment without requiring a temporary refitting of all devices.”

Comment:

2) The choice of frequency is also not clear. Such low frequencies can pass through walls. So even if a small 4 by 4 RIS rejects the interfering signals, the interfering signals can pass through the walls and reach the users. So it is not really clear what is the advantage here. This is very important because the authors compare their work with other works in Table 1 which are at higher frequencies over which EM signals have smaller penetration depths and implementing filters with higher Q are more difficult.

Response:

The comment is appreciated. We agree that it is necessary to note the reasoning behind choosing

the frequency for this study. We chose 3.5 GHz because it falls within the 4G LTE Band 42 (3.4-3.6 GHz) and the primary band for 5G technology^{[3][4]}. In our future work, the proposed RIS on a large scale will be implemented in existing wireless networks to evaluate its promising performance.

The reviewer points out that signals at this frequency can pass through walls, thus potentially bypassing the RIS and reaching the users. We apologize for any misunderstandings caused by Fig. 1 in our previous manuscript, which depicts a small RIS with only 1×4 subarrays installed on the wall. In a typical application scenario within a shielded room, a large-scale RIS covering the entire wall is necessary to effectively perform signal selection and channel customization^{[5]-[8]}. Fig. 1 is now revised in the manuscript, as shown in Fig. R1 in this response letter. In this work, to prove the concept, the small RIS with 1×4 subarrays is fabricated and measured.

We agree with the reviewer that electromagnetic (EM) waves at higher frequencies have smaller penetration through walls, especially at millimeter-wave (mmWave) frequencies. However, traditional walls are not designed to have the filtering performances as the proposed RIS. On the other hand, to compare the proposed technique with other RISs at higher frequencies, we have designed a mmWave filtering RIS using the same method, whose structure is shown in Fig. R2a. A filtering chip (MMCB2528G5T-0001A3, TDK)^[9] is employed here. We conduct field-circuit cosimulations to study the features of this structure, and the results are illustrated in Fig. R2b and 2c. The reflection amplitudes of the four coding states range from -3.3 to -6.3 dB between 26.4 GHz and 30.4 GHz. On the two sides of the passband, a 30-dB rejection in the stopbands is obtained. In the operating bandwidth, stable 90° phase differences are exhibited between the curves. The Q factor of this mmWave filtering RIS is 7.1.

The results are compared with Ref. [10] in Table R1. The operating bandwidth (BW) means the frequency range with the acceptable phase-shifting ability. $BW_n\text{dB}$ refers to the bandwidth in which the transmission amplitude is lower than the maximum value by less than n dB. It is observed that our $BW_{3\text{dB}}$ and $BW_{20\text{dB}}$ are narrower than the reference. The ratio $BW/BW_{3\text{dB}}$ is used to evaluate the operating BW as a percentage of $BW_{3\text{dB}}$. Compared with Ref. [10], our

mmWave RIS has a larger BW/BW3dB value, suggesting a narrower occupied bandwidth. The rectangle coefficient K20dB (defined by the ratio BW20dB/BW3dB) of our mmWave RIS is 1.4, indicating steeper transitions on the two edges of the transmission curve and thus a better filtering effect than the reference. These results prove the stronger filtering and phase-tuning properties of the proposed RIS in this manuscript.

Table R1. Comparison between the mmWave RIS in this work and the one in Ref. [10].

Ref.	Type	f_0 (GHz)	Operating BW (GHz)	BW3dB (GHz)	BW20dB (GHz)	BW/BW3dB	Q factor	K20dB
mmWave RIS (Simulated)	Rx. - Fun. - Tx.	28.5	3.6 (12.6%)	4.0 (14.0%)	5.6 (19.6%)	90.0%	7.1	1.4
[10]	Rx. - Tx.	32.0	4.3 (13.4%)	4.9 (15.3%)	10 (31.2%)	87.8%	6.5	2.0

f_0 , center frequency; BW, bandwidth; BW $_n$ dB, n dB bandwidth; Q factor, quality factor, the ratio f_0 /BW3dB; K20dB, rectangle coefficient, the ratio BW20dB/BW3dB; Rx., receiver; Fun., functional module; Tx., transmitter.

Corresponding changes:

1) In the third paragraph of the “Introduction” section, the scale of the RIS shown in Fig. 1 is described.

“A large-scale RIS is mounted on the wall indoors”

2) Fig. 1 and its caption in the manuscript are revised. They are the same as Fig. R1 in this response letter.

3) In the third paragraph of the “Design of Filtering RIS” section, we describe the practical scale of the RIS and the scale that was studied in this particular work.

“In practical application scenarios, the proposed filtering RIS would be implemented on a much larger scale, requiring a significant number of elements. In this study, we focus on a small panel that features 1×4 subarrays to demonstrate the feasibility of our proposed concept, as depicted in **Fig. 3.**”

4) In the fourth paragraph in the “Introduction” section, we explain the reason for the frequency selection.

“Its central frequency is 3.5 GHz with a 200 MHz passband, which falls in the 4G LTE Band

42 and the primary band for the 5G technology^{43, 44}.”

5) Details of the millimeter-wave filtering RIS are provided in Supplementary Information Note 6:

“Supplementary Note 6: Design of a millimeter-wave filtering RIS

To compare the proposed technique with other RISs at higher frequencies, we have designed a mmWave filtering RIS using the same method, whose structure is shown in **Fig. S7a**. A filtering chip (MMCB2528G5T-0001A3, TDK)^[7] is employed here. We conduct field-circuit cosimulations to study the features of this structure, and the results are illustrated in Figs. S7b and 7c. The reflection amplitudes of the four coding states range from -3.3 to -6.3 dB between 26.4 GHz and 30.4 GHz. On the two sides of the passband, a 30-dB rejection in the stopbands is obtained. In the operating bandwidth, stable 90° phase differences are exhibited between the curves. The Q factor for this mmWave filtering RIS is 7.1.

The results are compared with Ref. [8] in **Table S2**. Compared with Ref. [8], our mmWave RIS has a larger BW/BW3dB value. The rectangle coefficient K20dB of our mmWave RIS is 1.4, indicating steeper transitions on the two edges of the transmission curve and thus a better filtering effect than the reference. These results prove the stronger filtering and phase-tuning properties of the proposed RIS in this manuscript.”

Fig. S7 and Table S2 in the revised Supplementary Information are the same as Fig. R2 and Table R1 in this response letter, respectively.

Comment:

3) Please add a comment on how this device can be extended to 2D? It does not seem to be straightforward as the signal collected by each subarray is phased differently (in contrast to conventional RISs where each element is phased). This is important because figure 1 promises a 2D beamforming capabilities by the implemented device does not yield that.

Response:

The comment is appreciated. It is true that an RIS used in real-life scenarios should have its elements individually controlled for 2D beamforming capabilities, but it is important to note that this requires complex feeding networks and can be expensive to fabricate. As a result, only a limited number of current studies have conducted concept validations with all the elements individually controlled ^{[11]-[13]}. In most cases, instead, the elements were controlled in columns for the concept demonstration^{[1][2][14]-[19]}.

To demonstrate the concept proposed in this work, a small sample with 1×4 subarrays is studied through simulations and experiments. Each subarray consists of four elements that share a functional module. In other words, the four elements in a column are synchronized in operation, and thus the beam-steering property is illustrated in a specific azimuthal plane. Based on the validated mechanism, it is feasible to scale down the subarray and even allow each element to have its own functional module for filtering and phase shifting, thus achieving the 2D beamforming capabilities depicted in Fig. R1.

Corresponding change:

In the last paragraph of the “Design of Filtering RIS” section, we explain the reason for using the small sample for the concept demonstration. A method for extending the lattice into a 2D configuration is also presented.

“It is important to note that only four subarrays are used in this study as a proof of concept to demonstrate the functions of RIS. The elements are controlled in columns, which is consistent with most of the previous studies^{22, 32-34, 52-55}. However, based on the validated mechanism, it is feasible to reduce the scale of the subarray and equip each element with its own functional module for filtering and phase shifting. With an increase in the number of elements and their distribution into a large lattice, the advanced 2D beam-manipulating effects could be achieved.”

Comment:

4) Please provide a reference for the following sentence: "...however, their practical deployments are hindered by the limited frequency selectivity, which is described quantitatively by the quality factor (Q factor)...." To the best of my knowledge, that has not been a hindering factor. There are obstacles in using RISs in practice, but their low Q has not been an issue.

Response:

The comment is appreciated. It is indeed necessary to explicitly explain the significance of the frequency selectivity in RIS-assisted wireless communications.

In a recent publication on the electromagnetic (EM) interference in RIS-aided communications [20], the authors pointed out that “the common practice is to consider the signals generated by the system and thereby neglecting the electromagnetic interference (EMI) or “noise” (or “pollution”) that is inevitably present in any environment. The EMI may arise from a variety of natural, intentional, or non-intentional causes.” They believed that the EMI re-radiated by the RIS might “degrade the end-to-end SNR of the system.”

In Ref. [21], the authors stated that “the coexistence of multiple networks is a traditional problem in practical wireless mobile communication networks.” They suggested that “in actual networks, the wireless signals incident on an RIS include both the “target signals” coming from the network to which the RIS belongs and the “nontarget signals” coming from neighboring networks.” It was emphasized that one of the purposes of using RIS is to control the propagation of EM waves, but existing RISs usually have broadband tuning capabilities, so they may lead to serious network coexistence problems. In this work, the filtering mechanism was proposed to “reduce or eliminate the unexpected tuning of nontarget signals.”

Based on the above discussions, we can see that existing RISs lacking proper frequency selectivity are insufficient for meeting the demands of wireless mobile communication networks. To address this issue, we have proposed the filtering RIS with a high Q factor to not only effectively mitigate interference from adjacent spectra but also enhance signal propagation within the target frequency band.

Corresponding change:

In the second paragraph of the “Introduction” section, we highlighted the problem of practical wireless communication networks and the limitation of existing RISs with low Q factors.

“It should be noticed that the practical wireless mobile communication scenarios often have multiple networks occupying closely adjacent spectra. However, existing RISs typically have broadband properties that not only tune signals in the target spectrum but also affect nontarget signals, resulting in serious network coexistence problems or even security concerns^{35, 36}. In addition, most of current RISs failed to consider the impact of electromagnetic interference or noise, whether it is intentional or non-intentional³⁷. This lack of frequency selectivity, which can be measured in terms of a quality factor (Q-factor), hinders their practical deployment. This issue should be taken seriously in today’s wireless environment with the increasingly crowded spectrum.”

Comment:

5) What is the difference between this device and a repeater/relay? All seems to be the same except that repeaters/relays allow for amplification of the signal which is in contrast to the 50% loss of power offered by this device.

Response:

Thank you for the comment. It is indeed important to list the difference between the proposed RIS and a repeater/relay. There have been several pieces of literature talking about this issue^{[6][7]}. Compared with a relay, the advantages of the RIS are briefly discussed as follows.

1. As presented in Ref. [6], “the wireless environment is modeled as an exogenous entity that cannot be controlled, but only adapted to.” The utilization of relays is one of the common approaches to capitalize on the uncontrollable wireless environment. However, relays are active devices that need dedicated power sources for operation. They are equipped with active electronic components, such as analog-to-digital converters (ADCs), digital-to-analog converters (DACs), mixers, and power amplifiers for transmission. Therefore, the deployment of relays is costly and power-consuming.

In contrast, the RIS provides a promising solution to shape the wavefront of EM waves and thus make the wireless environment customizable. Different from a relay, it does not need the active components mentioned above. It is mainly composed of a dielectric substrate, metallic patches, and functional microstrip lines. Its reconfigurable properties are achieved by controlling the switching status of PIN diodes through simple direct-current (DC) biasing wires. Therefore, the RIS has a much lower power consumption and complexity. Only power supplies for the controlling module are required, resulting in significant energy savings.

2. As presented in Ref. [6], full-duplex (FD) relays introduce high loop-back self-interference because of the concurrent transmission and reception of signals. Additionally, they generate co-channel interference at the destination, since relays and transmitters emit different information on the same physical resource. On the contrary, the proposed RIS does not contain any non-reciprocal components and therefore supports an FD mode of operation at a very low cost.

3. The active electronic components used in relays are responsible for the presence of additive noise. In amplify-and-forward (AF) relaying, the noise is also amplified at the relays ^[6]. The RIS does not contain any amplifiers at the current stage, so its performances are not affected by the additive noise.

4. Ref. [22] demonstrates that the decode-and-forward (DF) relays are more sensitive to electromagnetic interference (EMI), which may arise from a variety of causes, e.g., other (single or multiple) transmitting devices and/or natural background radiation. The authors believed that “RIS-aided communications are more resilient to EMI” because of the spatial filtering capabilities of RISs. Beyond that, the proposed RIS provides the filtering effect in the frequency domain, providing a stronger ability to mitigate EMI.

On the other hand, the proposed RIS does show the unwanted loss in the passband. As analyzed in the response to Comment #7, the loss of the RIS is primarily due to the filter chip, which comes at the cost of the sharp frequency selection and high rejection in the stopbands. The insertion loss of the phase shifters and conversion loss between the spatial waves and the guided waves also contribute to the result. To address this issue, a possible solution would be to

integrate low-noise amplifiers in the functional modules of the RIS, while ensuring minimal noise is introduced to the signal, as predicted by Ref. [23].

Corresponding changes:

1) In the last paragraph of the “Introduction” section, we briefly compare the RIS and conventional repeaters/relays and emphasize the advantages of the RIS.

“Unlike the conventional repeaters or relays that contain active components like analog-to-digital/digital-to-analog converters, mixers, and power amplifiers, the proposed RIS has a significantly lower power consumption and complexity, and is free of additive noises⁴⁵. More discussions on the RIS and conventional repeaters or relays are provided in Supplementary Information Note 8.”

2) In the second paragraph of the “Fabrication and Measurement” section, the loss of the RIS is briefly analyzed, and the possible solution is raised.

“The loss is mainly due to the insertion loss of the phase shifters and filter chips, which comes at the cost of phase shifting and sharp frequency selection features. More details about the loss analysis are presented in Supplementary Information Note 5. To address this issue, low-noise amplifiers could be integrated into the functional modules to compensate for the attenuation.”

3) In Supplementary Information Note 8, More discussions on the RIS and conventional repeaters or relays are provided.

“Supplementary Note 8: Comparison with Conventional Repeater/Relays

Compared with a relay, the advantages of the RIS are briefly discussed as follows.

1. As presented in Ref. [9], “the wireless environment is modeled as an exogenous entity that cannot be controlled, but only adapted to.” The utilization of relays is one of the common approaches to capitalize on the uncontrollable wireless environment. However, relays are active devices that need dedicated power sources for operation. They are equipped with active electronic components, such as analog-to-digital converters (ADCs), digital-to-analog converters (DACs), mixers, and power amplifiers for transmission. Therefore, the deployment of relays is costly and power-consuming.

In contrast, the RIS provides a promising solution to shape the wavefront of EM waves and thus make the wireless environment customizable. Different from a relay, it does not need the active components mentioned above. It is mainly composed of a dielectric substrate, metallic patches, and functional microstrip lines. Its reconfigurable properties are achieved by controlling the switching status of PIN diodes through simple direct-current (DC) biasing wires. Therefore, the RIS has a much lower power consumption and complexity. Only power supplies for the controlling module are required, resulting in significant energy savings.

2. As presented in Ref. [9], full-duplex (FD) relays introduce high loop-back self-interference because of the concurrent transmission and reception of signals. Additionally, they generate co-channel interference at the destination, since relays and transmitters emit different information on the same physical resource. On the contrary, the proposed RIS does not contain any non-reciprocal components and therefore supports an FD mode of operation at a very low cost.

3. The active electronic components used in relays are responsible for the presence of additive noise. In amplify-and-forward (AF) relaying, the noise is also amplified at the relays ^[9]. The RIS does not contain any amplifiers at the current stage, so its performances are not affected by the additive noise.

4. Ref. [10] demonstrates that the decode-and-forward (DF) relays are more sensitive to electromagnetic interference (EMI), which may arise from a variety of causes, e.g., other (single or multiple) transmitting devices and/or natural background radiation. The authors believed that “RIS-aided communications are more resilient to EMI” because of the spatially filtering capabilities of RISs. Beyond that, the proposed RIS provides the filtering effect in the frequency domain, providing a stronger ability to mitigate EMI.

On the other hand, the proposed RIS does show the unwanted loss in the passband. As analyzed in Note 5, the loss is primarily due to the filter chip, which comes at the cost of the sharp frequency selection and high rejection in the stopbands. The insertion loss of the phase shifters

and conversion loss between the spatial waves and the guided waves also contribute to the result. To address this issue, a possible solution would be to integrate low-noise amplifiers in the functional modules of the RIS, while ensuring minimal noise is introduced to the signal, as predicted by Ref. [4].”

Comment:

6) I believe the proposed communication demonstration does not show interference mitigation which is the promise of the work. Instead of using absorbers around the device, the authors should use typical walls. On one side, place two sources, one with 3.5 GHz and one with 3.9 GHz. See the received signal on the other side for both and report SNIR with and without RIS.

Response:

Thank you for the constructive comments. We accepted the suggestion and redesigned the wireless communication experiments both indoors and with a real wall outside.

1) Indoor experiment

The scenarios of the indoor experiments and the measured results are provided in Fig. R3. The pictures of the experiments are given in Fig. R4. Two USRPs, which were set to be working at 3.5 and 3.9 GHz, respectively, encoded a color picture into a binary stream and modulated it using a quadrature phase shift keying (QPSK) scheme. Two pairs of horn antennas were employed. The transmitting antenna no. 1 and receiving antenna no. 1 were connected to the output and input ports of USRP 1 at 3.5 GHz; transmitting antenna no. 2 and receiving antenna no. 2 were connected to the ports of USRP 2 at 3.9 GHz. The transmitting and receiving antennas were placed on the two sides of a windowed absorbing screen.

Five cases were designed to validate the proposed concept. In Case 1, the window was left empty. In Case 2, a metallic plate of the same size as the metasurface was placed on the window. The two pairs of antennas faced normally at the absorbing screen. In Case 1, as expected, the demodulated constellation diagrams at the two frequencies were of high quality, and the pictures were satisfactorily recovered. In contrast, the presence of the metallic plate blocked the

transmissions, as shown in Fig. R4b. In Case 3, the metallic plate was replaced by the RIS sample, and the coding sequence on the RIS was set to be “0000”, indicating that the passband signal (3.5 GHz) should be transmitted through the RIS in a direction perpendicular to the surface. The receiving antennas were placed in the correct direction. As shown in Fig. R4c, the quality of the demodulated constellation diagram at 3.5 GHz was good, and the picture was satisfactorily restored, implying that the signal was well received. In Case 4, the coding sequence was changed to “0123”, directing the beam to a 28° angle on the yo z -plane. By repositioning the receiving antennas to the correct direction, almost the same satisfactory results were observed at 3.5 GHz, as presented in Fig. R4d. However, if the receiving antenna deviated from the expected direction in Case 5, the signal was no longer correctly received, as proved by the cluttered constellation diagram and the unrecovered picture in Fig. R4e. These results vividly demonstrate the wave-manipulation capability of the RIS in the passband. Contrary to these results, for the signals at 3.9 GHz, it is observed that no matter what the coding sequence was or the location of the receiving antenna on the right side, the picture could not be restored, and the constellation diagrams were chaotic, as shown in Fig. R4c through 4e. This strongly suggests that these signals were rejected by the RIS.

To further quantitatively evaluate the performance of the signal transmissions, we calculated the signal-to-noise ratio (SNR) spectra that varied with the radiated power from the transmitting antennas in the five cases, which are shown in Fig. R4f. It is observed that the SNR progressively increased with the rise in the radiated power. More importantly, the SNRs at 3.5 GHz in Cases 1, 3, and 4 were significantly higher than those in Cases 2 and 5, proving the beam-steering capability of the RIS in the passband. At 3.9 GHz, the SNRs in Cases 2 through 5 were considerably lower than those in Case 1, suggesting the blocking effect of the RIS in the stopband. Fig. R4g shows the SNR spectrum in Case 3 when the coding sequence was “0000” and the radiated power was 0 dBm, showing a frequency window that allows the signals to be transmitted through the RIS efficiently. These data were in good agreement with the phenomena presented in Fig. R4a through 4e.

2) Outdoor experiment

As a response to the comment, we replaced the absorbing screen with a brick wall and conducted the measurements outdoors in the five cases. The pictures are offered in Fig. R6. The thickness of the wall was about 23 mm. Before conducting the wireless communication experiments, we measured the transmission amplitudes through the whole brick wall. A vector network analyzer (VNA) (Agilent N5245A) was used for this purpose. Two horn antennas, which were connected to port 1 and port 2 of the VNA, were located on the two sides of the wall. The results are shown in Fig. R5a, showing that the attenuation caused by the whole wall was around 10 dB, much lower than the absorbing screen. Then, a window was created on the wall with the same size as the RIS, and the RIS was placed in the window. The RIS was coded by the sequence “0000”. The transmission through the wall was measured again, and the results are shown in Fig. R5a. Due to the small size of the RIS and the relatively strong transmission of the wall, the bandpass of the RIS can hardly be distinguished by the curve. Because this is not a standard measurement procedure for transmission performances, the results are not rigorous enough to reflect the true frequency response of the proposed RIS, which is shown in Fig. R5b.

We still performed the wireless communication experiments. The setups were the same as those indoors except that the absorbing screen was replaced by the wall. In Fig. R6, the transmitting antennas were on the other side of the wall, so they were not shown. From the results in Cases 1 and 2, almost identical phenomena were observed with or without the metallic plate. This is due to the poor shielding effect of the wall and the small size of the plate. Then, we put the RIS in the window in Cases 3, 4, and 5. From the results in Fig. R6c through 6e, the blocking effect on the 3.9 GHz signals can be barely observed. At 3.5 GHz, no matter how we moved the receiving antenna or adjusted the coding sequence, the signals could still be received and recovered well. Fig. R7 presents the measured SNR spectra versus the radiated power in the five cases, which are in accordance with the phenomena in Fig. R5a. The introduction of the RIS did not cause any significant change to the SNR values. As shown in Fig. R8 about the SNR spectrum versus the frequency with the radiated power of 0 dBm, it is difficult to tell the filtering effect of the RIS on the communication performances. These results are quite different

from those with the absorbing screen, and they are attributed to the small size of the RIS and the relatively strong transmission property of the wall.

Nevertheless, we believe that the novel performances of the RIS have been effectively demonstrated by the small array. In our experiments, uncontrollable multipath signals were effectively suppressed by the absorbing screen around the RIS, which follows the standard measurement for microwave devices in laboratories ^{[5][8]}.

Corresponding changes:

1) The indoor wireless communication experiments, which were redesigned according to the comment, are presented on Pages 10 through 11 of the revised manuscript.

“The wireless communication configuration and measured results are presented in **Fig. 6** and **Fig. 7**. Two software-defined radio reconfigurable devices (USRP-2943, National Instruments Corp.)²⁵⁻²⁷ that are set to work at 3.5 and 3.9 GHz, respectively, encode a color picture into a binary stream and modulate it using a quadrature phase shift keying (QPSK) scheme. Two transmitting antennas and two receiving antennas are connected to the output and input of the two USRPs, respectively. Horn antennas with a working bandwidth that covers the entire frequency band of interest (3.1-3.9 GHz) are employed here to eliminate the uncontrollable multipath effects. A windowed absorbing screen is placed between the transmitting and receiving antennas, and the RIS is embedded in the window. The transmitting antennas are placed facing the RIS in a normal orientation, and the distance between the antennas and RIS is 1 meter. The positions of the receiving antennas are varied on the *yo**z*-plane. The QPSK signals generated by USRPs are radiated by the antennas and pass through the RIS before being received by the other set of antennas. The received signals are then demodulated by the USRPs to recover the pictures. The symbol rate in experiments is set as 200 KBaud, which is enough for picture transmissions.

Five different cases are designed to showcase the effectiveness of the filtering RIS, as shown in Figs. 7a through 7e. The radiated power from the transmitting antennas is 0 dBm. Settings of the RIS and directions of the receiving antennas are summarized in **Table 2**. Photographs of

the measurements can be found in Supplementary Information Note 7. Cases 1 and 2 serve as the control groups. In Case 1, the absorption window with the same size as the RIS is left empty, while in Case 2, a metallic plate is placed inside it. As depicted in Fig. 7a, the demodulated constellation diagrams at both 3.5 GHz and 3.9 GHz are of high quality, and the pictures are satisfactorily recovered, implying that the signals are well received by the antenna through the window. In Case 2, on the contrary, the presence of a metal plate prevents the transmission, as shown in Fig. 7b. In Case 3, the metallic plate is replaced by the RIS, and the coding sequence on it is set to “0000”, indicating that the passband signal (3.5 GHz) should be transmitted through the RIS in a direction perpendicular to the surface. The receiving antennas are placed in the correct direction. As shown in Fig. 7c, the demodulated constellation diagram at 3.5 GHz is of good quality, and the picture is satisfactorily restored, implying that the signal is well received. In Case 4, the coding sequence is changed to “0123”, directing the beam to a 28° angle on the *yo*z-plane. By repositioning the 3.5-GHz receiving antenna to the correct direction, almost the same satisfactory results are observed, as presented in Fig. 7d. However, if the 3.5-GHz receiving antenna deviates from the expected direction (Case 5), the signal is no longer correctly received, as proved by the cluttered constellation diagram and the unrecovered picture in Fig. 7e. The experiments vividly demonstrate the wave-manipulation capability of the RIS in the passband. In sharp contrast, for the out-of-passband signal (3.9 GHz), regardless of the coding sequence or the receiving antenna’s location on the right side, the picture cannot be restored, and the constellation diagram is cluttered. This strongly suggests that such signals are rejected by the RIS.

To further quantitatively evaluate the performance of the signal transmissions, we calculate the signal-to-noise ratio (SNR) spectrum in relation to the radiated power from the transmitting antenna in the five cases, as shown in Fig. 7f. It is observed that the SNR progressively increases with the rise in the radiated power, indicating an enhancement in transmission through the RIS. For the passband signal (3.5 GHz), the SNRs in Cases 1, 3, and 4 are significantly higher than those in Cases 2 and 5, proving the beam-steering capability of the RIS in the passband. For the out-of-band signal (3.9 GHz), the SNRs in Cases 2 through 5 are considerably lower than those

in Case 1, suggesting the blocking effect of the RIS in the stopband. Fig. 7g shows the SNR spectrum in Case 3 using the horn antennas when the coding sequence is “0000” and the radiated power is 0 dBm. It can be seen that the maximum value of 25.7 dB occurs at 3.5 GHz. The value exceeds 23.2 dB from 3.4 to 3.6 GHz, and it drops sharply below 11.6 dB outside the 3.3 to 3.7 GHz range, indicating a frequency window that allows the signals to be transmitted through the RIS efficiently. These results demonstrate the wave-manipulation capability of the RIS in the passband and interference mitigation for adjacent frequencies, aligning with the results presented in Figs. 7a through 7e.”

Table 2. Settings of the RIS and receiving antennas in the wireless communication experiments.

Cases	Coding sequences on the RIS	Theoretical beam direction	Receiving antenna direction
1	Air	-	0°
2	PEC	-	All
3	0000	0°	0°
4	0123	28°	28°
5	0123	28°	Any except 28°

2) Fig. 7 in the manuscript is revised, which is the same as Fig. R3 in this response letter.

3) Descriptions and pictures of the indoor wireless communication experiments are provided in Supplementary Information Note 7.

“**Fig. S8** illustrates the constellation diagrams, recovered pictures, and experiment photographs of the five cases in the wireless communication measurement. Horn antennas are employed as the transmitting and receiving antennas to eliminate uncontrollable multipath effects. The constellation diagrams in Cases 3 and 4 are cluttered, and the pictures are not well recovered.”

Fig. S8 in the revised Supplementary Information are the same as Fig. R4 in this response letter.

Comment:

7) Can this work be used at higher frequencies? If yes, how can the losses be kept at minimum?
The 50% loss of power in this work is not small.

Response:

Thank you very much for the comment. The RIS in this work was specifically designed with the passband centered at 3.5 GHz, as demonstrated in the manuscript, so it cannot be directly applied at higher frequencies. However, theoretically speaking, the same concept and design method can be extended to higher frequencies. To do this, the structures involving the receivers, transmitters, and phase shifters should be reduced because the wavelength decreases. Additionally, an appropriate filter chip operating at the target frequency should be selected carefully. Here, a new RIS working at mmWave frequencies was designed to prove the concept. Please refer to the response to Comment #2 for the design of the RIS and the simulation results, showing much better performances than the current report ^[10].

The main source of the RIS losses is analyzed as follows.

1. Conversion loss between the spatial waves and the guided waves. According to the data provided in Supplementary Information Note 2, the conversion efficiency of the 4×4 receiver/transmitter is 95%-97%, which means that the conversion loss of the whole panel is between 6% and 10%. This loss can be reduced by optimizing the radiation properties of the receiver and transmitter and improving the impedance match between the receiver/transmitter and the circuits.

2. Insertion loss of the filter chips. As depicted in Fig. S4 in Supplementary Information, the filter chip achieves the stopbands with over 30 dB rejection, but it also has an insertion loss of 1.5-2.3 dB. This implies that about 29%-42% of the power is attenuated.

3. Insertion loss of the phase shifters. According to the data presented in Fig. S6a of Supplementary Information, the loss is around 0.7-1.0 dB, meaning that about 15%-21% of the power is attenuated. It is primarily caused by the ohmic loss of the PIN diode.

The above discussions reveal that the loss caused by the filter chip contributes significantly to the RIS's overall insertion loss. It occurs at the expense of the sharp frequency selection and high rejection in the stopbands. This can be mitigated by using low-loss filter chips. Moreover, the loss caused by the phase shifters can be reduced by selecting PIN diodes with superior switching characteristics. Inspired by the current amplifying RISs [23]-[25], we can further integrate low-noise amplifiers into the functional module to compensate for the loss.

Corresponding changes:

1) In the last paragraph of the “Design of Filtering RIS” section, the method to move the operating frequency of the RIS is discussed.

“Additionally, this RIS is designed with the passband centered at 3.5 GHz, but this concept can be moved to higher frequencies by adjusting the structural parameters accordingly and selecting an appropriate filter chip that operates at the desired target frequencies. In Supplementary Information Note 6, we have designed a filtering RIS that operates at the millimeter-wave (mmWave) frequencies using the same concept. Its filtering and phase-shifting properties are studied through field-circuit cosimulations. As shown in Table I, its performances are competitive when compared to the state-of-the-art work in the mmWave bands.”

2) Details of the millimeter-wave filtering RIS are provided in Supplementary Information Note 6:

“Supplementary Note 6: Design of a millimeter-wave filtering RIS

To compare the proposed technique with other RISs at higher frequencies, we have designed a mmWave filtering RIS using the same method, whose structure is shown in **Fig. S7a**. A filtering chip (MMCB2528G5T-0001A3, TDK)^[7] is employed here. We conduct field-circuit cosimulations to study the features of this structure, and the results are illustrated in Figs. S7b and 7c. The reflection amplitudes of the four coding states range from -3.3 to -6.3 dB between 26.4 GHz and 30.4 GHz. On the two sides of the passband, a 30-dB rejection in the stopbands is obtained. In the operating bandwidth, stable 90° phase differences are exhibited between the curves. The Q factor for this mmWave filtering RIS is 7.1.

The results are compared with Ref. [8] in **Table S2**. Compared with Ref. [8], our mmWave RIS has a larger BW/BW_{3dB} value. The rectangle coefficient K_{20dB} of our mmWave RIS is 1.4, indicating steeper transitions on the two edges of the transmission curve and thus a better filtering effect than the reference. These results prove the stronger filtering and phase-tuning properties of the proposed RIS in this manuscript.”

Fig. S7 and Table S2 in the revised Supplementary Information are the same as Fig. R2 and Table R1 in this response letter, respectively.

3) In the second paragraph of the “Fabrication and Measurement” section, the loss of the RIS is briefly analyzed, and the possible solution is raised.

“The loss is mainly due to the insertion loss of the phase shifters and filter chips, which comes at the cost of phase shifting and sharp frequency selection features. More details about the loss analysis are presented in Supplementary Information Note 5. To address this issue, low-noise amplifiers could be integrated into the functional modules to compensate for the attenuation.”

4) In Supplementary Information Note 5, the loss of the RIS is detailed, and the possible solution is raised.

“**Supplementary Note 5: Loss Analysis**”

The main source of the RIS losses is analyzed as follows.

1. Conversion loss between the spatial waves and the guided waves. According to Fig. S3, the conversion efficiency of the 4×4 receiver/transmitter is 95%-97%, which means that the conversion loss of the whole panel is between 6% and 10%. This loss can be reduced by optimizing the radiation properties of the receiver and transmitter and improving the impedance match between the receiver/transmitter and the circuits.

2. Insertion loss of the filter chips. As depicted in Fig. S4, the filter chip achieves the stopbands with over 30 dB rejection, but it also has an insertion loss of 1.5-2.3 dB. This implies that about 29%-42% of the power is attenuated.

3. Insertion loss of the phase shifters. According to Fig. S6a, the loss is around 0.7-1.0 dB, meaning that about 15%-21% of the power is attenuated. It is primarily caused by the ohmic loss of the PIN diode.

The above discussions reveal that the loss caused by the filter chip contributes significantly to the RIS's overall insertion loss. It occurs at the expense of the sharp frequency selection and high rejection in the stopbands. This can be mitigated by using low-loss filter chips. Moreover, the loss caused by the phase shifters can be reduced by selecting PIN diodes with superior switching characteristics. Inspired by the current amplifying RIS^{[4]-[6]}, we can further integrate low-noise amplifiers into the functional module to compensate for the loss.”

Comment:

8) Please comment on what communication link in practice uses horn antennas for communication at 3.5 GHz band? This is a very important question because spatial selectivity of high gain antennas already provides with some level of interference mitigation.

Response:

Thank you for the valuable comment. Indeed, high-gain horn antennas have a high level of directivity in the spatial domain. However, the purpose of our experiments was to validate the filtering property of the proposed metasurface in the frequency domain. The horn antennas' operating bandwidths are wider than the spectrum of our interest, so their impact in the frequency domain can be neglected. In the spatial domain, to ensure that our demonstration was not degraded by uncontrollable multipath effects, we utilized the horns and measured the transmission coefficients and far-field patterns in a microwave anechoic chamber. The horn antennas were positioned several meters away from the metasurface to provide plane wave excitations. As a matter of fact, horn antennas are commonly employed in antenna measurements to eliminate the influence of multipath signals. In previous works involving RISs and their performances in wireless communications, horn antennas were utilized as plane-wave excitations [1], [2], [13], [26], [27].

As a response to the reviewer's comment, we have carried out the wireless communication

experiments using two pairs of custom-built patch antennas. As shown in Fig. R9, one pair of the patch antennas operate at 3.5 GHz with a gain of 5.5 dBi, and the other pair work at 3.9 GHz with a gain of 5.9 GHz. Compared with the horn antennas, the directivities of the patch antennas are lower. The experiment setups are shown in Fig. R10, which were almost the same as those in Fig. R4 except that the horn antennas were replaced by the patch antennas.

Fig. R10 illustrates the constellation diagrams, recovered pictures, and experiment photographs of the five cases in the wireless communication measurement using the patch antennas. Similar to the results in Fig. R4, the signals were received and recovered in Cases 1, 3 and 4 at the 3.5 GHz, as shown in Fig. R10a, 10c and 10d. The signals could not be received in Cases 2 and 5 at the 3.5 GHz. This is because the metallic plate was placed in the window in Case 2, and in Case 5, the beam was directed at a 28° angle, but the receiving antenna was not aligned at the same 28° angle, as demonstrated in Fig. R10b and 10e. For the 3.9 GHz, the signals could only be received and recovered in Case 1, where nothing was placed in the window, as shown in Fig. R10a.

Fig. R11a and 11b present the SNR versus radiated power in the five cases at 3.5 and 3.9 GHz, respectively. The frequency-selecting and beam-steering performances of the proposed RIS could be distinctly identified. First, at 3.5 GHz in Fig. R11a, the SNR values in Cases 3 and 4 were larger than those in Case 5, verifying the beamforming feature of the RIS. Second, as shown in Fig. R11c, the SNR values in Cases 3 and 4 were higher at 3.5 GHz than those at 3.9 GHz, demonstrating the filtering effect of the RIS.

However, we observe that the differences in SNR between Case 1 (Air) and Case 2 (PEC) were approximately 10 dB, which was notably smaller than the 20 dB difference reported in the manuscript using horn antennas. We attribute this discrepancy to the influence of multipath effects. Furthermore, the measurement of the SNR spectrum was not practical due to the limited bandwidths of the patch antennas, which is illustrated in Fig. 7(g) in the manuscript.

Corresponding changes:

1) In the second paragraph on Page 10 of the revised manuscript, the reason for using the horn

antennas is added.

“Horn antennas with a working bandwidth that covers the entire frequency band of interest (3.1-3.9 GHz) are employed here to eliminate the uncontrollable multipath effects.”

2) In the last paragraph of the “Fabrication and Measurement” section in the revised manuscript, the wireless communication experiments employing the patch antennas are added.

“The wireless communication experiments are also conducted using custom-built patch antennas as both transmitting and receiving antennas. Their directivities are lower than those of the horns. The photographs and the results can be found in Supplementary Information Note 7. The conclusions drawn from these experiments are consistent with those obtained using the horn antennas.”

3) In Supplementary Information Note 7, the wireless communication experiments employing patch antennas are added.

“**Fig. S10** depicts the same wireless communication setup as that in Fig. S8, except that the horn antennas are replaced by two pairs of custom-built patch antennas. One pair of patch antennas operate at 3.5 GHz, and the other pair work at 3.9 GHz. Their gains are 5.5 dBi and 5.9 dBi, respectively, and the input reflection coefficients at their working frequencies are less than -15 dB, as shown in **Fig. S9**. **Fig. S11** presents the SNR versus the radiated power in the five cases at 3.5 and 3.9 GHz. At 3.5 GHz in Fig. S11a, the SNR values in Cases 3 and 4 are larger than those in Case 5, confirming the beamforming feature of the RIS. In Fig. S11c, the SNR values in Cases 3 and 4 are higher at 3.5 GHz than those at 3.9 GHz, demonstrating the filtering effect of the RIS. The experiments demonstrate the impressive frequency-selecting and beam-steering characteristics of the filtering RIS. ”

Comment:

9) It is not clear why authors refer to generalized Snell's law in discussion for (1). This is a typical receiving/transmitting antenna. Its operation can be described using antenna array theory.

Response:

Thank you for the comment. It is indeed true that the far-field scattering patterns of the metasurface can be described using the antenna array theory, which is calculated by [28]:

$$, \quad (R1)$$

where M and N are numbers of elements in the x - and y -directions; $F(\theta, \phi)$ is the transmission far-field pattern of the element (m, n) ; φ_{mn} is the transmission phase of the element (m, n) ; k is the wavenumber of the EM wave in free space; d is the period of the receiver and transmitter elements; θ and ϕ are the elevation and azimuthal angles, respectively.

Corresponding change:

In the first paragraph on Page 8 of the revised manuscript, we describe the far-field scattering patterns of the metasurface using the antenna array theory.

“These patterns agree quite well with the theoretical results presented in Fig. 4e, which are calculated by ^[1]

$$, \quad (1)$$

where M and N are numbers of elements in the x - and y -directions; $F(\theta, \phi)$ is the transmission far-field pattern of the element (m, n) ; φ_{mn} is the transmission phase of the element (m, n) ; k is the wavenumber of the EM wave in free space; d is the period of the receiver and transmitter elements; θ and ϕ are the elevation and azimuthal angles, respectively.”

References:

- [1] Usman, M. et al. Intelligent wireless walls for contactless in-home monitoring. *Light Sci. Appl.* **11**, 212 (2022).
- [2] Rains, J. High-Resolution Programmable Scattering for Wireless Coverage Enhancement: An Indoor Field Trial Campaign. *IEEE Trans. Antennas Propag.* **71**, 518–530 (2023).
- [3] Website of 4G LTE Networks, <https://www.4g-lte.net/about/lte-frequency-bands/lte-band-42/>.
- [4] “5G Spectrum Public Policy Position” in white paper, (2017).
- [5] Chen, W., Wen, C.-K., Li, X. & Jin, S. Channel Customization for Joint Tx-RISs-Rx Design in Hybrid mmWave Systems. *IEEE Transactions on Wireless Communications* **22**, 8304–8319 (2023).
- [6] Di Renzo, M. Reconfigurable Intelligent Surfaces vs. Relaying: Differences, Similarities, and Performance Comparison. *IEEE Open Journal of the Communications Society* **1**, 798–807 (2020).
- [7] Bjornson, E., Ozdogan, O. & Larsson, E. G.. Intelligent Reflecting Surface Versus Decode-and-Forward: How Large Surfaces are Needed to Beat Relaying?. *IEEE Wireless Communications Letters* **9**, 244–248 (2020).
- [8] Di Renzo, M. Smart Radio Environments Empowered by Reconfigurable Intelligent Surfaces: How it Works, State of Research, and the Road Ahead. *IEEE Journal on Selected Areas in Communications* **38**, 2450–2525 (2020).
- [9] https://product.tdk.com/en/search/rf/rf/filter/info?part_no=MMCB2528G5T-0001A3.
- [10] Cheng, C.-C. & Abbaspour-Tamijani, A. Study of 2-bit antenna–filter–antenna elements for reconfigurable millimeter-wave lens arrays. *IEEE Trans. Microw. Theory Tech.* **54**, 4498–4506 (2006).
- [11] Wang, H. P. et al. Noncontact Electromagnetic Wireless Recognition for Prosthesis Based on Intelligent Metasurface. *Adv. Sci.* **9**, e2105056, (2022).
- [12] Zhao, H. et al. Metasurface-assisted massive backscatter wireless communication with commodity Wi-Fi signals. *Nat. Commun.* **11**, 3926, (2020).
- [13] Li, W. et al. Intelligent metasurface system for automatic tracking of moving targets and

- wireless communications based on computer vision. *Nat. Commun.* **14**, 989, (2023).
- [14]Hu, Q. et al. An Intelligent Programmable Omni-Metasurface. *Laser & Photonics Reviews*, **17**, 2100718 (2022).
- [15]Tang, W. et al. Wireless Communications With Reconfigurable Intelligent Surface: Path Loss Modeling and Experimental Measurement. *IEEE Transactions on Wireless Communications* **20**, 421-439, (2021).
- [16]Ren, S. et al. Configuring Intelligent Reflecting Surface With Performance Guarantees: Blind Beamforming. *IEEE Transactions on Wireless Communications* **22**, 3355-3370, (2023).
- [17]Tang, W. et al. Path Loss Modeling and Measurements for Reconfigurable Intelligent Surfaces in the Millimeter-Wave Frequency Band, arXiv:2101.08607, (2021)
- [18]Pei, X. et al. RIS-Aided Wireless Communications: Prototyping, Adaptive Beamforming, and Indoor/Outdoor Field Trials. *IEEE Transactions on Communications* **69**, 8627-8640, (2021).
- [19]Araghi, A. et al. Reconfigurable Intelligent Surface (RIS) in the Sub-6 GHz Band: Design, Implementation, and Real-World Demonstration. *IEEE Access* **10**, 2646-2655, (2022).
- [20]De Jesus Torres, A., Sanguinetti, L. & Bjornson, E. Electromagnetic Interference in RIS-Aided Communications. *IEEE Wireless Communications Letters* **11**, 668–672 (2022).
- [21]Zhao, Y. & Lv, X. Network Coexistence Analysis of RIS-Assisted Wireless Communications. *IEEE Access* **10**, 63442-63454, (2022).
- [22]Torres, A., Sanguinetti, L., Björnson, E. Intelligent Reconfigurable Surfaces vs. Decode-and-Forward: What is the Impact of Electromagnetic Interference? arXiv:2203.08046.
- [23]Wang, X., et al. Amplification and Manipulation of Nonlinear Electromagnetic Waves and Enhanced Nonreciprocity using Transmissive Space-Time-Coding Metasurface, *Adv. Sci.*, **9**, 2105960 (2022).
- [24]Ma, Q. et al. Controllable and Programmable Nonreciprocity Based on Detachable Digital Coding Metasurface. *Advanced Optical Materials* **7**, (2019).
- [25]Wu, L., et al. A Wideband Amplifying Reconfigurable Intelligent Surface. *IEEE Transactions on Antennas and Propagation* **70**, 10623–10631 (2022).

- [26]Zhang, L. et al. A wireless communication scheme based on space- and frequency-division multiplexing using digital metasurfaces. *Nature Electronics* **4**, 218-227, (2021).
- [27]Tang, W.. On Channel Reciprocity in Reconfigurable Intelligent Surface Assisted Wireless Networks. *IEEE Wireless Communications* **28**, 94–101 (2021).
- [28]Cui, T. J., Qi, M. Q., Wan, X., Zhao, J. & Cheng, Q. Coding metamaterials, digital metamaterials and programmable metamaterials. *Light Sci. Appl.* **3**, e218–e218 (2014).

Fig. R1 A typical application scenario of the proposed filtering RIS, which is on a large scale and covers the entire wall of a shielded room. The base stations, named BS_1 , BS_2 , and BS_3 , are located outdoors working at three adjacent frequencies f_1 , f_2 , and f_3 , respectively. The filtering RIS placed indoors on the wall aims to enhance the quality of the wireless communications between the base station BS_2 and the indoor users IU_1 , IU_2 , and IU_3 by generating specific pencil beams and collimating them accurately toward the targets. Different from the conventional RISs, the filtering RIS exhibits a powerful frequency-selecting ability that allows only the f_2 signal to enter the room, but strongly rejects the outdoor f_1 and f_3 signals. Hence, potential interference issues caused by the out-of-band signals can be eliminated.

Fig. R2 (a) Configuration of the millimeter-wave filtering RIS. The transmission coefficients of the millimeter-wave filtering RIS obtained from the cosimulations. (b) Amplitude and (c) phase spectra of the four digital states (States 0-3).

Fig. R3 The wireless communication experiments to demonstrate the proposed selectivity in both frequency and spatial domains of the filtering RIS. (a)-(e) Five cases with different frequencies and coding sequences. (f) The SNR spectra in relation to the radiated power from the transmitting antennas in the five cases. (g) The SNR spectrum in Case 3 with the radiated power of 0 dBm.

Fig. R4 Pictures of the five cases in the realistic wireless communication scenario using the absorbing screen.

Fig. R5 (a) Transmission amplitudes of the whole brick wall and the wall with the RIS embedded in the window. (b) The transmission amplitude of the absorbing screen with the RIS embedded in the window. The RIS was coded by the sequence “0000”.

Fig. R6 Five cases in the realistic wireless communication scenario where a brick wall was used.

Fig. R7 The SNR spectra in relation to the radiated power in five cases, where a brick wall was used, at (a) 3.5 GHz and (b) 3.9 GHz, and (c) in Cases 3, 4 at 3.5 GHz and 3.9 GHz.

Fig. R8 The SNR spectrum in Case 3 where the RIS was coded as “0000” and the brick wall was used.

Fig. R9 Details of the patch antennas used in the wireless communication experiments. (a) Input reflection coefficients. (b) Measured gains. (c) Photographs of the patch antennas working at 3.5 and 3.9 GHz, respectively.

Fig. R10 Five cases in the realistic wireless communication scenario using the patch antennas.

Fig. R11 The SNR spectra as a function of the radiated power using the patch antennas in the five cases at (a) 3.5 GHz and (b) 3.9 GHz. (c) SNR in Cases 3 and 4 with the radiated power of 0 dBm.

REVIEWER COMMENTS

Reviewer #1 (Remarks to the Author):

Authors replied to all the raised comments in satisfactory manner and I have no more comments

Reviewer #2 (Remarks to the Author):

The authors have addressed most of my concerns. Their detailed modifications are appreciated. However, there are still some important changes required:

1) Figure R6 needs to be added to the manuscript or the supplemental material. It captures the salient features of the proposed device in a more practical setting.

2) Furthermore, the pros and cons of the proposed device over frequency selective surfaces need to be discussed. FSSs provide frequency interference mitigation over many bands and can be easily implemented over the whole walls or windows with almost no cost and complexity compared to the complicated proposed devices. For example, see the following manuscripts:

Sung, Grace Hui-hsia, Kevin W. Sowerby, Michael J. Neve, and Allan G. Williamson. "A frequency-selective wall for interference reduction in wireless indoor environments." *IEEE Antennas and Propagation Magazine* 48, no. 5 (2006): 29-37.

Sung, Grace Hui-hsia, Kevin W. Sowerby, Michael J. Neve, and Allan G. Williamson. "A frequency-selective wall for interference reduction in wireless indoor environments." *IEEE Antennas and Propagation Magazine* 48, no. 5 (2006): 29-37.

The following paper also shows a passband at 3.5 GHz.

Shi, Yongrong, Wanchun Tang, Wei Zhuang, and Cheng Wang. "Miniaturised frequency selective surface based on 2.5-dimensional closed loop." *Electronics letters* 50, no. 23 (2014): 1656-1658.

Yang, Guohui, Tong Zhang, Wanlu Li, and Qun Wu. "A novel stable miniaturized frequency selective surface." *IEEE Antennas and Wireless Propagation Letters* 9 (2010): 1018-1021.

At the proposed frequency, the beam redirection is not crucial as most indoor wireless systems operating at these frequencies do not need beamforming or directive beams. Furthermore, the authors have not shown or discussed a viable way to add 2D beamforming to the proposed device. In other words, it seems that the primary improvement of the proposed device over FSSs is 1D beamforming. A detailed comparison with FSSs literature is needed.

3) In almost all studies, the incident signal is considered normal to the RIS. What happens if the transmitter is at angle from this device? This is also related to the previous comments as FSSs can be designed to support both polarizations and accept/reject signals from all incident angles.

RE: Manuscript No. NCOMMS-23-39119A, “A filtering reconfigurable intelligent surface for interference-free wireless communications”

RESPONSES TO REVIEWERS’ COMMENTS

The authors would like to express their sincere gratitude to the editors and reviewers for their invaluable time and diligent efforts on this manuscript. We have taken all the comments seriously and made every effort to respond point by point. All corresponding changes are highlighted in the revised manuscript. We sincerely hope that the comments and suggestions have been addressed adequately.

Reviewer #1:

Comment:

Authors replied to all the raised comments in satisfactory manner and I have no more comments

Response:

Thank you very much. Your encouragement is appreciated.

Reviewer #2:

Comment:

The authors have addressed most of my concerns. Their detailed modifications are appreciated. However, there are still some important changes required:

Response:

Thank you for the positive feedback. The comments have been carefully responded to, and corresponding changes have been made in the manuscript.

Comment:

1) Figure R1 needs to be added to the manuscript or the supplemental material. It captures the salient features of the proposed device in a more practical setting.

Response:

Thank you for the constructive comment. We accept the suggestion and have added Fig. R1 as Fig. S18 in Supplementary Note 9, where the descriptions of the figure are also presented.

Fig. R1 Five cases in the realistic wireless communication scenario where the absorbing screen around the RIS is replaced with a brick wall. The two transmitting horn antennas are on the other side of the wall, hence not shown in the pictures.

Corresponding changes:

1) At the end of the “Fabrication and Measurement” section, the outdoor wireless communication experiments are mentioned.

“Two additional wireless communication experiments are conducted to further illustrate the properties of the RIS. The first experiment is carried out by using the custom-built patch antennas as both transmitting and receiving antennas, and the second experiment is conducted outdoors by replacing the absorbing screen around the RIS with a brick wall. The photographs and results are given in Supplementary Information Note 9.”

2) Details of the outdoor wireless communication experiments are provided in Supplementary Information Note 9.

“We also perform wireless communication experiments outdoors where the absorbing screen around the RIS is replaced with a brick wall. The thickness of the wall is about 23 cm. The pictures are offered in **Fig. S18**. The transmitting and receiving antennas are still the horn antennas, which are placed on the two sides of the wall. It should be noticed that almost identical results are obtained with or without the metallic plate (Cases 1 and 2, respectively), which is attributed to the poor shielding effect of the wall and the small size of the plate. Then, we put the RIS in the window in Cases 3, 4, and 5, and the results are given in Figs. S18c through 18e. The blocking effect of the RIS on the 3.9 GHz signals can be barely observed. At 3.5 GHz, no matter how we move the receiving antenna or adjust the coding sequence, the signals can still be received and recovered well. These results are quite different from those with the absorbing screen, and they are attributed to the small size of the RIS and the relatively strong transmission property of the wall. It is believed that the performance will improve by using an RIS with a larger aperture.”

3) Fig. S18 in the revised Supplementary Information is the same as Fig. R1 in this response letter.

Comment:

2) Furthermore, the pros and cons of the proposed device over frequency selective surfaces need to be discussed. FSSs provide frequency interference mitigation over many bands and can be easily implemented over the whole walls or windows with almost no cost and complexity compared to the complicated proposed devices. For example, see the following manuscripts:

Sung, Grace Hui-hsia, Kevin W. Sowerby, Michael J. Neve, and Allan G. Williamson. "A frequency-selective wall for interference reduction in wireless indoor environments." *IEEE Antennas and Propagation Magazine* 48, no. 5 (2006): 29-37.

Sung, Grace Hui-hsia, Kevin W. Sowerby, Michael J. Neve, and Allan G. Williamson. "A frequency-selective wall for interference reduction in wireless indoor environments." *IEEE Antennas and Propagation Magazine* 48, no. 5 (2006): 29-37.

The following paper also shows a passband at 3.5 GHz.

Shi, Yongrong, Wanchun Tang, Wei Zhuang, and Cheng Wang. "Miniaturised frequency selective surface based on 2.5-dimensional closed loop." *Electronics letters* 50, no. 23 (2014): 1656-1658.

Yang, Guohui, Tong Zhang, Wanlu Li, and Qun Wu. "A novel stable miniaturized frequency selective surface." *IEEE Antennas and Wireless Propagation Letters* 9 (2010): 1018-1021.

At the proposed frequency, the beam redirection is not crucial as most indoor wireless systems operating at these frequencies do not need beamforming or directive beams.

Furthermore, the authors have not shown or discussed a viable way to add 2D beamforming to the proposed device. In other words, it seems that the primary improvement of the proposed device over FSSs is 1D beamforming. A detailed comparison with FSSs literature is needed.

Response:

The valuable comment is highly appreciated. The proposed filtering RIS is compared with the FSSs reported in previous studies ^{[1]-[5]}, including the ones suggested by the reviewer ^{[1][4][5]}. As shown in Table R1, the FSSs in Refs. [4] and [5] were designed with the bandstop property, functioning in the opposite way to the proposed bandpass filtering RIS that allows the center frequency to pass through.

It is true that the FSSs are less expensive, but two main advantages are demonstrated by the proposed RIS. Firstly, it shows a much stronger frequency selectivity via a single-layer configuration. This can be proved by the comparison made in the table, showing that the K20dB value of the RIS is almost the same as that of the stacked 5-layer FSSs ^{[2][3]}, but the RIS outperforms them in terms of BW3dB, BW20dB, and Q factor.

The second advantage held by the RIS over the FSSs is the ability for beam manipulation. We respectfully disagree with the reviewer who commented that the beam redirection indoors is not important. As discussed in Refs. [6-8], RISs have shown significant potential in indoor wireless systems. Compared to the passive FSSs, the beamforming function of the RIS can optimize the wireless channel, thereby increasing the signal strengths in target directions and potentially enhancing the interference immunity for indoor wireless communications. Additionally, it is believed that RISs on interior walls are of great help in circumventing indoor blockages and extending coverage of cell signals in the sub-6G frequency bands ^{[9][10]}. For example, an average enhancement of 16 dB to the received signal was achieved from 3 to 4.5 GHz in the indoor field trial, which was facilitated by the programmable beam-steering behaviors ^[9]. In Ref. [10], by reconfiguring the RIS working at 3.5 GHz to align with a

specific direction of interest, the signal level at the receiver side was enhanced by more than 15 dB.

Moreover, we stress that the improvement of the proposal over the FSSs is not just the 1D beamforming illustrated in the manuscript. Based on the proposed design and the preliminary performance illustrations, it is practical to develop an RIS structure capable of realizing 2D beam manipulations. To briefly prove this, we designed a new filtering RIS element following the same principle, which has a pair of transmitter and receiver and a functional module for filtering and phase shifting. The structure of the element and the structural parameters are shown in Figs. R2a and 2b, respectively, and its simulated transmission amplitudes and phases are shown in Figs. R2c and 2d, respectively. The transmission amplitudes of the four phase coding states range from -1.2 to -4.2 dB between 3.4 and 3.6 GHz. On the two sides of the passband, rejections of 30 dB in the stopbands are obtained. In this operating bandwidth, stable 90° phase differences are exhibited between the curves. The Q factor of this filtering RIS element is 14.4, and the K20dB is 1.4, which are extremely close to the performances of the subarray discussed in the manuscript.

To verify the 2D beamforming ability, we constructed an RIS with 10×10 elements and simulated the far-field patterns of transmitted signals when it is normally illuminated by plane waves at 3.5 GHz. The results are illustrated in Fig. R5, showing that by carefully adjusting the phases of the elements, the beam can be tilted within wide angle ranges of $\pm 63^\circ$ and $\pm 64^\circ$ on the *xoz*- and *yo_z*-planes, respectively. The viable way to realize the 2D beamforming using the proposed technique is thus proved.

On the other hand, the shortcomings of the proposed RIS compared with the FSSs are the single polarization and the relatively narrow incident angle scope under the transverse-electric (TE) mode illumination. This will be discussed in detail in the response to Comment #3.

Table R1. Comparison between the proposed RIS and the FSSs in previous studies.

Ref.	Configuration	Filtering type	f_0 (GHz)	BW3dB (GHz)	BW20dB (GHz)	Q factor	K20dB
This work	Rx. - Fun. - Tx.	Bandpass	3.5	0.25 (7.1%)	0.32 (9.1%)	14	1.3
[1]	Single Layer FSS	Bandpass	3.82	0.5 (13.1%)	>2.5 (65.4%)	7.64	>5
[2]	Stacked 5-Layers	Bandpass	5.4	0.9 (16.7%)	1.1 (20.4%)	6.0	1.2
[3]	Stacked 5-Layers	Bandpass	5.4	0.7 (13.0%)	0.9 (16.7%)	7.7	1.3
[4]	Single Layer FSS	Bandstop	5.8	-	-	-	-
[5]	Single Layer FSS	Bandstop	2.85	-	-	-	-

f_0 , center frequency; BW n dB, n dB bandwidth; Q factor, quality factor, the ratio f_0 /BW3dB; K20dB, rectangle coefficient, the ratio BW20dB/BW3dB; Rx., receiver; Fun., functional module; Tx., transmitter.

Fig. R2 (a) Configuration of the newly designed single filtering RIS element. (b)

Structural parameters of the element. $P_x=41.39$ mm, $P_y=40$ mm, $H_1=1.0$ mm, $H_2=2.0$ mm, $W_1=18.78$ mm, $W_2=23.38$ mm, $W_3=20.56$ mm, $W_4=7.53$ mm, $W_5=2.45$ mm, $L_1=5.21$ mm, $L_2=10.03$ mm, $L_3=11.41$ mm, $L_4=10.03$ mm, $R_1=1.52$ mm, $R_2=1.72$ mm.

(c) and (d) The simulated transmission coefficients of the filtering RIS element. (c) Amplitude and (d) phase spectra of the four phase coding states (States 0-3) under the normal incidence.

Fig. R3 The simulated transmission properties of the newly designed single RIS element. (a) The amplitude spectra with Coding state 0 under the TM and TE mode with incident angles from 0° to 60°. Transmission (b) amplitudes and (c) phases at 3.5 GHz versus the oblique incident angles with the four coding states.

Fig. R4 The Q factor and K20dB of the newly designed single filtering RIS element versus the incident angles under (a) TM and (b) TE modes.

Fig. R5 The simulated far-field patterns of the RIS with 10×10 newly designed single elements. Each RIS element has a pair of transmitter and receiver and a functional module for filtering and phase shifting. (a) xoz -plane. (b) yoz -plane.

Corresponding changes:

1) Comparison between the proposed RIS and the FSSs is described in the last paragraph of the “Introduction” section.

“Compared with the conventional frequency selective surfaces (FSSs) that have been widely employed for frequency selectivity⁴⁵⁻⁴⁹, the proposed single-layer RIS has the advantage of significantly improved filtering performance. Beyond that, the proposal’s flexible beamforming capability makes it more useful for increasing the signal strengths in target directions and enhancing the interference immunity for indoor wireless communications^{30, 34, 50-52}. Detailed comparisons between the RISs and FSSs are provided in Supplementary Information Note 10.”

2) In Supplementary Information Note 10, discussions on the RIS and FSSs are provided.

“Supplementary Note 10: Comparison between the proposed RIS and frequency selective surfaces (FSSs)

Generally, there are two types of FSSs, bandpass FSSs^{[9]-[11]} and bandstop FSSs^{[12][13]}. The bandpass FSSs allow signals around the center frequency to pass through, while the bandstop FSSs operate oppositely. Here the proposed RIS is compared with the bandpass FSSs.

Two main advantages are demonstrated by the proposed RIS over the bandpass FSSs. Firstly, it shows a much stronger frequency selectivity via a single-layer configuration. This can be proved by the comparison made in Table S3, showing that the K20dB value of the RIS is almost the same as that of the stacked 5-layer FSSs, but the RIS outperforms them in terms of BW3dB, BW20dB, and Q factor.

The second advantage held by the proposed RIS over the FSSs is the ability for beam manipulation. As discussed in references ^[14-16], RISs have shown significant potential in indoor wireless systems. Compared to the passive FSSs, the beamforming function of the RIS can optimize the wireless channel, thereby increasing the signal strengths in target directions and potentially enhancing the interference immunity for indoor wireless communications. Additionally, it is believed that RISs on interior walls are of great help in circumventing indoor blockages and extending coverage of cell signals in the sub-6G frequency bands ^[14-18].”

Table S3 in the revised Supplementary Information is the same as Table R1 in this response letter.

3) In the last paragraph of the “Design of Filtering RIS” section, the new RIS element and the 2D beamforming verification are briefly described.

“Based on the validated mechanism, it is feasible to design a filtering RIS with a pair of transmitter and receiver and a functional module for filtering and phase shifting, which can realize the two-dimensional (2D) beam-steering performance. In Supplementary Information Note 7, the design of the element and its simulated properties are presented. Through simulations, the 2D beamforming is also verified by using an array with 10×10 elements, showing the wide scanning ranges of $\pm 63^\circ$ and $\pm 64^\circ$ on the xoz - and $yo z$ -planes, respectively.”

4) Details of the filtering RIS that can realize the 2D beamforming are provided in Supplementary Information Note 7.

“Supplementary Note 7: Design of the Single RIS Element and 2D Beamforming

We have designed a new filtering RIS element following the same principle, which has a pair of transmitter and receiver and a functional module for filtering and phase shifting. The structure of the element and the structural parameters are shown in **Figs. S9a** and **9b**, and its simulated transmission amplitudes and phases are shown in **Figs. S9c** and **9d**, respectively. The transmission amplitudes of the four phase coding states range from -1.2 to -4.2 dB between 3.4 and 3.6 GHz. On the two sides of the passband, rejections of 30 dB in the stopbands are obtained. In this operating bandwidth, stable 90° phase differences are exhibited between the curves. The Q factor of this filtering RIS element is 14.4, and the K20dB is 1.4. which are extremely close to the results of the subarray discussed in the main text.

We further studied the transmission coefficients of this RIS element under TM and TE oblique incidence ranging from 0° to 60°. As examples, the amplitude spectra with Coding state 0 are depicted in **Fig. S10a**. It is observed that the amplitude spectra are minimally affected by the incident angle. The results with the other three coding states are similar. The passband frequency remains stable, and rejections of more than 20 dB are observed outside the passband. We then focused on the transmission amplitudes and phases at 3.5 GHz that vary with incident angles for the four coding states, as shown in **Figs. S10b** and **10c**, respectively. The amplitudes under the TM incidences are quite stable even when the angle is as large as 60°; the amplitudes under the TE incidences start to deteriorate when the angle is about 30°, and end up to about -5 dB when the angle is 60°. The Q factors and K20dB values are shown in **Figs. S11a** and **11b**, respectively. It is observed that the Q factors are higher than 14 and the K20dB values remain almost unchanged in all conditions, indicating stable filtering responses under both the TM and TE oblique incidences. For the transmission phase performances, stable 90° shiftings are exhibited under both TM and TE incidences, as displayed in **Fig. S10c**.

To verify the 2D beamforming ability, we constructed an RIS with 10×10 elements and

simulated the far-field patterns of transmitted signals when it is normally illuminated by plane waves at 3.5 GHz. The results are illustrated in Fig. S12, showing that by carefully adjusting the phases of the elements, the beam can be tilted within wide angle ranges of $\pm 63^\circ$ and $\pm 64^\circ$ on the xoz - and $yo z$ -planes, respectively. The viable way to realize the 2D beamforming using the proposed technique is thus proved.”

Figs. S9, S10, S11 and S12 in the revised Supplementary Information are the same as Figs. R2, R3, R4 and R5 in this response letter, respectively.

Comment:

3) In almost all studies, the incident signal is considered normal to the RIS. What happens if the transmitter is at angle from this device? This is also related to the previous comments as FSSs can be designed to support both polarizations and accept/reject signals from all incident angles.

Response:

Thank you for the comment. The FSSs in the suggested references indeed exhibited outstanding wide-angle properties for both polarizations. We conducted a series of simulations and measurements to study the transmission coefficients of the proposed RIS under oblique incidences. The amplitude spectra under the transverse-electric (TM) mode with incidence angles ranging from 0° to 60° are presented for starters. As examples, the simulated and measured results with Coding state 0 are depicted in Figs. R6a and 6b, respectively, which agree quite well with each other. As the incident angle increases, the passband frequency remains stable, and rejections of more than 20 dB are obtained outside the passband. The results with the other three coding states are similar. Under the TM incidence, the transmission amplitudes and phases at 3.5 GHz that vary with the incident angles for the four coding states are shown in Figs. R6c and 6d, respectively. It can be seen that the amplitudes vary between -2 and -4 dB. The phase differences between the four coding states are nearly 90° . These results indicate the stable performances of the proposed RIS under TM oblique incidences.

Under the TE oblique incidence, both the filtering performance and the transmission amplitude degrade as the incident angle increases to 20° , as shown in Figs. R7a and 7b. This is because the in-phase condition of the four elements in the subarray deteriorates under the TE oblique incidence. The amplitudes and phases of the transmission coefficient versus the incidence angle at 3.5 GHz are shown in Figs. R7c and 7d, respectively. Putting aside the amplitude decline, the phase difference between the four coding states remains 90° as the incident angle varies.

We need to point out that the newly designed single RIS element, which is introduced earlier in this letter, features notably improved performances under the TE oblique incidences. Simulations are carried out under the TM and TE incidences with incident angles ranging from 0° to 60° . The amplitude spectra with Coding state 0 are depicted in Fig. R3a. It is observed that the amplitude spectra are minimally affected by the incident angle. The results with the other three coding states are similar. The passbands remain stable, and rejections of more than 20 dB are observed outside the passbands. We then focused on the transmission amplitudes at 3.5 GHz that vary with incident angles for the four coding states, as shown in Fig. R3b. The amplitudes under the TM incidence are quite stable even when the angle is as large as 60° . Under TE incidence, the amplitudes start to deteriorate when the angle is about 30° , and end up to about -5 dB when the angle is 60° . This is quite better than the performances of the subarray. The Q factors and K20dB values are shown in Figs. R4a and 4b, respectively. It is observed that the Q factors are higher than 14 and the K20dB values are around 1.3 in all conditions, indicating stable filtering responses under both the TM and TE oblique incidences. For the transmission phase performances, stable 90° shiftings are exhibited under both TM and TE incidences, as displayed in Fig. R3c.

The filtering RIS discussed in this article operates under a single polarization due to the transmitters/receivers used in the elements supporting only one polarization. However, the design of the RIS is highly flexible, as the receiver, functional module, and

transmitter can be separately designed based on the receiver-transmitter architecture. This means that a polarization-independent filtering RIS can be designed by using dual-polarization transmitters/receivers. This strategy has been demonstrated in previous pieces of literature. For example, fractal metallic patterns were utilized as transmitters and receivers for a miniature FSS design under both polarizations ^[11]; in Ref. [12], a dual-polarized tightly coupled dipole element was adopted in electromagnetic functional surface designs.

Fig. R6 (a) Simulated and (b) measured transmission amplitude spectra with Coding state 0 under the TM mode with incident angles from 0° to 60°. Transmission (c) amplitudes and (d) phases at 3.5 GHz versus the oblique incident angles with the four coding states.

Fig. R7 (a) Simulated and (b) measured transmission amplitude spectra with Coding state 0 under the TE mode with incident angles from 0° to 20°. Transmission (c) amplitudes and (d) phases at 3.5 GHz versus the oblique incident angles with the four coding states.

Corresponding changes:

1) In the second paragraph of the “Fabrication and Measurement” section, the performances under oblique incidence are described.

“The performances under oblique incidences are discussed in detail in Supplementary Information Note 6.”

2) In the fourth paragraph of the “Design of Filtering RIS” section, the extension to support both polarizations is described.

“Thirdly, although the filtering RIS is currently designed to support a single polarization, the polarization-independent properties can be practically realized by adopting the dual-polarized transmitter and receiver in the element^{57, 58.}”

3) Details of the performances of the filtering RIS under oblique incidences are

provided in Supplementary Information Note 6.

“Supplementary Note 6: Performances of the Subarray under Oblique Incidences

Simulations and measurements are conducted to analyze the transmission features of the proposed RIS under oblique incidences. The amplitude spectra under the transverse-magnetic (TM) mode with incidence angles ranging from 0° to 60° are studied for starters. The simulated and measured results with Coding state 0 are depicted in **Figs. S7a** and **7b**, respectively, which agree quite well with each other. As the incident angle increases, the passband frequency remains stable, with the transmission amplitudes larger than those in the stopbands by 20 dB. The results with the other three coding states are similar. Under the TM incidence, the transmission amplitudes and phases at 3.5 GHz that vary with the incident angles for the four coding states are shown in **Figs. S7c** and **7d**, respectively. It can be seen that the amplitudes vary between -2 and -4 dB. The phase differences between the four coding states are nearly 90° . These results indicate the stable performances of the proposed RIS under TM oblique incidences.

Under the transverse-electric (TE) oblique incidence, both the filtering performance and the transmission amplitude degrade as the incident angle increases to 20° , as shown in **Figs. S8a** and **8b**, respectively. This is because the in-phase condition of the four elements in the subarray deteriorates under the TE oblique incidence. The amplitudes and phases of the transmission coefficient versus the incidence angle at 3.5 GHz are shown in **Figs. S8c** and **8d**, respectively. Putting aside the amplitude decline, the phase difference between the four coding states remains 90° as the incident angle varies.”

Figs. S7 and **S8** in the revised Supplementary Information are the same as **Figs. R6** and **R7** in this response letter, respectively.

References:

- [1] Yang, G. et al. A novel stable miniaturized frequency selective surface, *IEEE Antennas and Wireless Propagation Letters* **9**, 1018-1021 (2010).

- [2] Pan, W., Huang, C., Chen, P., Pu, M., Ma, X., & Luo, X. A beam steering horn antenna using active frequency selective surface. *IEEE Trans. Antennas Propag.* **61**, 6218–6223 (2013).
- [3] Reis, J. R. et al. FSS-inspired transmitarray for two-dimensional antenna beamsteering. *IEEE Trans. Antennas Propag.* **64**, 2197–2206 (2016).
- [4] Sung, G. et al. A frequency-selective wall for interference reduction in wireless indoor environments, *IEEE Antennas and Propagation Magazine*, **48**, 29-37 (2006).
- [5] Shi, Y. et al. Miniaturised frequency selective surface based on 2.5-dimensional closed loop. *Electronics letters* **50**, 1656-1658 (2014).
- [6] Poulakis, M. Metamaterials Could Solve One of 6G’s Big Problems [Industry View]. *Proceedings of the IEEE* **110**, 1151-1158, (2022).
- [7] Di Renzo, M. et al. Smart Radio Environments Empowered by Reconfigurable Intelligent Surfaces: How It Works, State of Research, and The Road Ahead. *IEEE Journal on Selected Areas in Communications* **38**, 2450-2525, (2020).
- [8] Björnson, E. et al. Reconfigurable intelligent surfaces: A signal processing perspective with wireless applications. *IEEE Access* **10**, 2646 - 2655 (2021).
- [9] Rains, J. High-Resolution Programmable Scattering for Wireless Coverage Enhancement: An Indoor Field Trial Campaign. *IEEE Trans. Antennas Propag.*, **71**, 518–530 (2023).
- [10] Araghi, A. et al. Reconfigurable Intelligent Surface (RIS) in the Sub-6 GHz Band: Design, Implementation, and Real-World Demonstration. *IEEE Access.* **10**, 2169-3536 (2022).
- [11] Shufeng, Z. et al. Analysis of Miniature Frequency Selective Surfaces Based on Fractal Antenna–Filter–Antenna Arrays. *IEEE Antennas and Wireless Propagation Letters* **11**, 240-243, (2012).
- [12] Wang, B., Wu, W., Zong, Z.-Y., Sima, B.-Y. & Fang, D.-G. Electromagnetic functional surfaces related to frequency response control using back-loaded radio frequency circuits. *IEEE Trans. Antennas Propag.* **70**, 9425–9434 (2022).

REVIEWERS' COMMENTS

Reviewer #2 (Remarks to the Author):

The authors have addressed my concerns.